# Respiratory tissue-associated commensal bacteria offer therapeutic potential against pneumococcal colonization

Soner Yildiz[1†], João P Pereira Bonifacio Lopes[1†], Matthieu Bergé[1],
Víctor González-Ruiz[2,3], Damian Baud[1], Joachim Kloehn[1], Inês Boal-Carvalho[1],
Olivier P Schaeren[4,5], Michael Schotsaert[6], Lucy J Hathaway[4], Serge Rudaz[2,3],
Patrick H Viollier[1], Siegfried Hapfelmeier[4], Patrice Francois[1], Mirco Schmolke[1]*

[1]Department of Microbiology and Molecular Medicine, Faculty of Medicine, University of Geneva, Geneva, Switzerland; [2]Analytical Sciences, School of Pharmaceutical Sciences, University of Geneva, University of Lausanne, Geneva, Switzerland; [3]Swiss Centre for Applied Human Toxicology, Basel, Switzerland; [4]Institute for Infectious Disease (IFIK), University of Bern, Bern, Switzerland; [5]Graduate School GCB, University of Bern, Bern, Switzerland; [6]Department of Microbiology, Icahn School of Medicine at Mount Sinai, New York, United States

*For correspondence:
mirco.schmolke@unige.ch

†These authors contributed equally to this work

Competing interests: The authors declare that no competing interests exist.

**Abstract** Under eubiotic conditions commensal microbes are known to provide a competitive barrier against invading bacterial pathogens in the intestinal tract, on the skin or on the vaginal mucosa. Here, we evaluate the role of lung microbiota in Pneumococcus colonization of the lungs. In eubiosis, the lungs of mice were dominantly colonized by *Lactobacillus murinus*. Differential analysis of 16S rRNA gene sequencing or *L. murinus*-specific qPCR of DNA from total organ homogenates *vs.* broncho alveolar lavages implicated tight association of these bacteria with the host tissue. Pure *L. murinus* conditioned culture medium inhibited growth and reduced the extension of pneumococcal chains. Growth inhibition in vitro was likely dependent on *L. murinus*-produced lactic acid, since pH neutralization of the conditioned medium aborted the antibacterial effect. Finally, we demonstrate that *L. murinus* provides a barrier against pneumococcal colonization in a respiratory dysbiosis model after an influenza A virus infection, when added therapeutically.

## Introduction

Mucosal surfaces are major entry ports for microbial pathogens. In the densely colonized gut, commensal bacteria confer a prominent protective role against invading bacterial pathogens, in part by posing a competitive threshold within this ecological niche (*Barthel et al., 2003*). This biological barrier is absent in axenic mice and reduced in mice harboring a low-complexity microbiota colonized mice. Barrier function of the microbiota is consequently sensitive to antibiotics treatment in colonized conventionally housed mice (*Barthel et al., 2003*; *Brugiroux et al., 2017*). Skin microbiota poses a similar barrier against colonization by bacterial skin pathogens (reviewed in *Parlet et al., 2019*). In recent years, it became evident that the lower respiratory tract (LRT), previously considered quasi-sterile, also hosts a bacterial microbiota under healthy conditions (*Dickson et al., 2015*). Whether the bacterial LRT microbiota confers colonization resistance against bacterial pathogens, e.g. causing pneumonia, is unclear.

Recently, a number of groups, including ours, reported compositional changes in the intestinal and lower respiratory tract microbiota over the course of an IAV infection using a C57BL/6J mouse

model housed under specific pathogen-free (SPF) conditions (*Deriu et al., 2016*; *Groves et al., 2018*; *Planet et al., 2016*; *Yildiz et al., 2018*). This opened a temporarily limited window for *Salmonella enterica* serovar Typhimurium infection in the intestine, similarly as in mice orally treated with antibiotics (*Deriu et al., 2016*; *Yildiz et al., 2018*). We found only little qualitative or quantitative differences in respiratory microbiota of IAV infected mice by 16S rRNA gene-specific next generation sequencing and qPCR. Nevertheless, we observed enhanced secondary colonization with *Streptococcus pneumoniae* following IAV infections, as described previously (*McCullers and Rehg, 2002*). The outcome of these super-infections is determined by host factors, viral factors and factors of the bacterial pathogen (reviewed in *McCullers, 2014*).

IAV infections take place in the poly-microbial environment of the respiratory tract and bacterial colonization of the lung is important to prime alveolar macrophages (*Abt et al., 2012*). However, the immediate role of the lung microbiota composition and complexity on invading bacterial pathogens is poorly investigated. We recently discovered a high content of Lactobacillaceae in the lung microbiota of healthy SPF housed mice, which was not reported before (*Yildiz et al., 2018*). Here we identify these commensal bacteria as *Lactobacillus murinus*, most probably a strain of the altered Schaedler flora (ASF). We provide in vitro evidence that it can limit pneumococcal growth, e.g. through release of lactic acid and reduces of chain formation of pneumococcus. Importantly, therapeutic application of *L. murinus* after IAV infection reduced the burden of secondary pneumococcal pneumonia.

## Results

### *L. murinus* (ASF361) could be a major constituent of mouse lung microbiota in laboratory settings

We previously demonstrated dominance of Lactobacillaceae in total lower respiratory tract (LRT) homogenates of SPF housed C57Bl/6J mice by 16S rRNA gene-specific NGS (*Yildiz et al., 2018*). In order to characterize these Lactobacillaceae in greater detail, we independently isolated and characterized two clones of phenotypically dominant bacteria from LRT homogenates cultivated on Columbia agar plates. After whole genome sequencing and de novo assembly we determined the evolutionary relationships of the two isolates based on the 16S rRNA gene. The phylogeny presented in *Figure 1A* is based on the alignment of approximately 1400 nucleotides of the 16S rRNA gene. The lung-isolated strains clustered with *L. murinus* and were clearly separated from *L. animalis* and *L. apodemi* isolates. One likely source for *L. murinus* is the altered Schaedler flora (ASF), introduced in 1978 to standardize mouse model colonization (*Dewhirst et al., 1999*; *Schaedler et al., 1965*; *Wymore Brand et al., 2015*) (indicated as *L. murinus* Schaedler in *Figure 1B*). Blasting of the de novo assembled genome against available genomes revealed that indeed *L. murinus* ASF361 was the best match to our *L. murinus* strain isolate (*Figure 1B*, z-score = 1, PRJNA591640 and *Supplementary file 1*). Alignment of the whole genome of our Lactobacillus isolate to the recently published reference genome of *L. murinus* ASF361 (*Wannemuehler et al., 2014*) resulted in >99. 5% sequence identity (*Figure 1C*).

The high prevalence of *L. murinus* in the murine LRT microbiota was not reported by other studies. Previous analyses of murine LRT microbiota by 16S rRNA gene NGS revealed a higher diversity with a balanced distribution of Actinobacteria, Bacteroidetes, Firmicutes and Proteobacteria (*Barfod et al., 2013*; *Poroyko et al., 2015*; *Singh et al., 2017*; *Yadava et al., 2016*). We speculated that differences in sampling techniques (bronchoalveolar lavage fluid (BALF) *vs.* partial organs *vs.* total organs) might introduce a potential sampling bias by avoiding or limiting tissue-associated bacteria. To resolve this discrepancy, we directly compared the compositional profile of LRT microbiota from bronchoalveloar lavage fluid (BALF), total organ after bronchoalveloar lavage or total organ without manipulation from SPF (individually ventilated cage) and conventionally (open cage) housed C57Bl/6J mice (see methods section for details). As demonstrated before (*Yildiz et al., 2018*), we found predominantly Lactobacillaceae in total organ samples (*Figure 2A/B*). However, this family was poorly represented in BALF samples, indicating a tight association with lung tissue. Accordingly, alpha diversity was higher in BALF samples (*Figure 2C*) and the overall homogeneous composition of BALF microbiota confirmed previous analysis by other groups. Of note, total 16S copy numbers by qPCR on the same amount of input DNA were substantially higher in tissue-derived samples

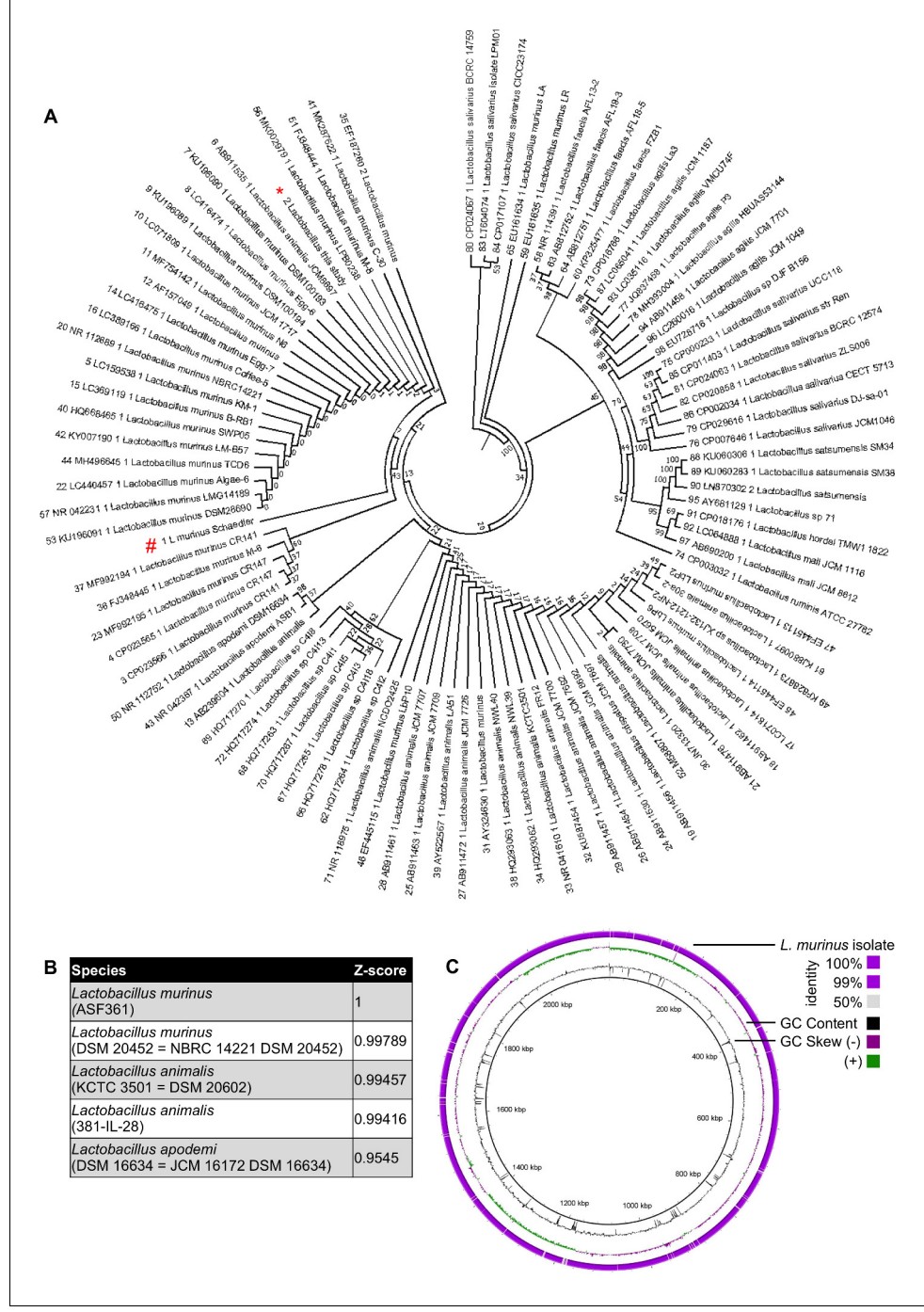

**Figure 1.** Lactobacillaceae isolate from total lung homogenates of mice is identified to be *Lactobacillus murinus*, a member of altered Scheadler Flora. (**A**) Phylogenetic tree of Lactobacillaceae isolate (indicated with red asterisk) to other known members of *Lactobacillaceae* family (*L. murinus* ASF361 is indicated with a red octothorpe). The phylogeny presented is based on the alignment of approximately 1400 nucleotides of the 16S rRNA gene. The phylogenetic analyses were generated with the neighbor-joining method. The percentage of replicate trees in which the associated taxa clustered together in the bootstrap test (100 replicates) is shown next to the branches. The trees are not rooted but drawn to scale with branch lengths in the same units as those of the evolutionary distances used to infer the phylogenetic tree. The evolutionary distances were computed using the Kimura 2-parameter method for 16S rRNA gene. The analysis included 17 sequences. Evolutionary analyses were conducted using MEGA6. All sequences are labeled according to strain name, species and accession number.( **B**) Genome of Lactobacillaceae isolate was aligned to reference genome database. Z-scores are calculated based on Average

*Figure 1 continued on next page*

*Figure 1 continued*

nucleotide identity. Top five hits are represented on the table. (**C**) Alignment between genome of *Lactobacillaceae* isolate and reference *L. murinus* genome from ASF 361. Genomic location, GC content, GC skew and sequence similarity given from most inner circle to most outer circle, respectively. The GC skew should not be considered since the nucleotide 0, origin of the genome, is not at the correct position due to the lack of closed genome of *L. murinus*.

---

(*Figure 2—figure supplement 1A*) than in BALF samples. Effectively, BALF samples were indistinguishable from blanks and background by 16S rRNA-specific qPCR. As a control for successful BAL procedure we included a qPCR for host 18S rRNA (*Figure 2—figure supplement 1B*) would be.

Regardless, the more sensitive NGS approach revealed OTUs in BALF (and tissue) samples, which were absent in blanks, such as Bacteroidia (*Figure 2A*). Beta-diversity (Weighted UniFrac) analysis revealed that BALF samples were found to be distinct in bacterial composition and abundance from tissue homogenates, resembling largely blank samples in SPF housed and separately positioned in CONV housed mice (*Figure 2D*). We next confirmed the presence of *L. murinus* in the lungs of SPF mice by two independent and species-specific techniques. First, we quantified *L. murinus* by specific primers using qPCR (*Figure 3A*). Genomic DNA from a defined amount of colony forming units of isolated *L. murinus* served as a standard to convert Ct values into cfu equivalents (*Figure 3—figure supplement 1A*). Four out of five SPF mice displayed DNA equivalent to >10e7 cfu/lung (*Figure 3A* and *Figure 3—figure supplement 1B*), which was about 100–1000 fold higher than the titers of life *L. murinus* determined by plating of lung homogenates (*Yildiz et al., 2018*) and potentially a result of dead *L. murinus*. In line with the 16S rRNA NGS date (*Figure 2A*) we did not detect *L. murinus* in DNA of BALF samples by *L. murinus*-specific qPCR, but in the majority of total lung DNA and DNA from lungs after BAL (*Figure 3B*). To gain insight into the anatomic distribution of *L. murinus* in the airways of mice, we performed highly sensitive RNAScope based ISH from frontal sections of total PFA fixed lungs (150 µm depth from ventral side) using probes specific for eubacteria or *L. murinus*. Bright red staining for *L. murinus* was exclusively visible in the airway space of SPF mouse lungs, but not in the germfree control specimen (*Figure 3C*). *L. murinus* was predominately found in large airways of the bronchi, but not in the trachea or the alveolar space (upper panels). Staining of adjacent slides for eubacteria (lower panels), revealed overall good correlation of *L. murinus*-specific staining with total eubacterial staining. This confirms our initial finding, that *L. murinus* dominates the bacterial flora. The RNAScope data further support that *L. murinus* is viable in the lung, since we detected short lived bacterial RNA. A limitation to these findings is that they are based on data from two laboratories within a single animal facility. Facility and vendor-specific confounding effects will require to be addressed in the future.

## *L. murinus* inhibits growth of bacterial lung pathogens

Direct competition of commensal microbiota with invading bacterial pathogens or already present pathobionts is believed to be a major host defense strategy against bacterial infections on mucosal surfaces. It was demonstrated for the intestine (*Barthel et al., 2003*; *Brugiroux et al., 2017*) and for the skin (*Nakatsuji et al., 2017*), but data from the LRT are currently not available. Mechanistically, commensal bacteria control outgrowth of pathobionts or invasion of pathogenic bacteria by different mechanisms: i.e. direct competition for nutrients and space, priming of innate immune cells, which are responsible for bacterial clearance and secretion of production/secretion of antibacterial metabolites (reviewed in *Pickard et al., 2017*).

Members of the family Lactobacillaceae were previously shown to inhibit growth of Gram-positive or Gram-negative bacteria in co-culture or by exposure to Lactobacillus conditioned medium (*De Keersmaecker et al., 2006*; *Lu et al., 2009*). Following a similar idea, we decided to first evaluate the effect of our *L. murinus* isolate on a common bacterial pathogen of the respiratory tract, *S. pneumoniae*, in an in vitro setting. Incompatibility of media requirements for efficient growth of these two bacteria did not allow us to perform co-culture studies. Instead, we used an experimental setup where *S. pneumoniae* was grown in presence of fresh MRS medium (FM) or *L. murinus* conditioned MRS medium (*Lm*CM), both at a dilution of 1:10 with *S. pneumoniae* culture media (TSB). To further rule out that the antibacterial effects could merely be a consequence of depletion/reduction of nutrients in the growth media, we used a similarly prepared medium conditioned by control

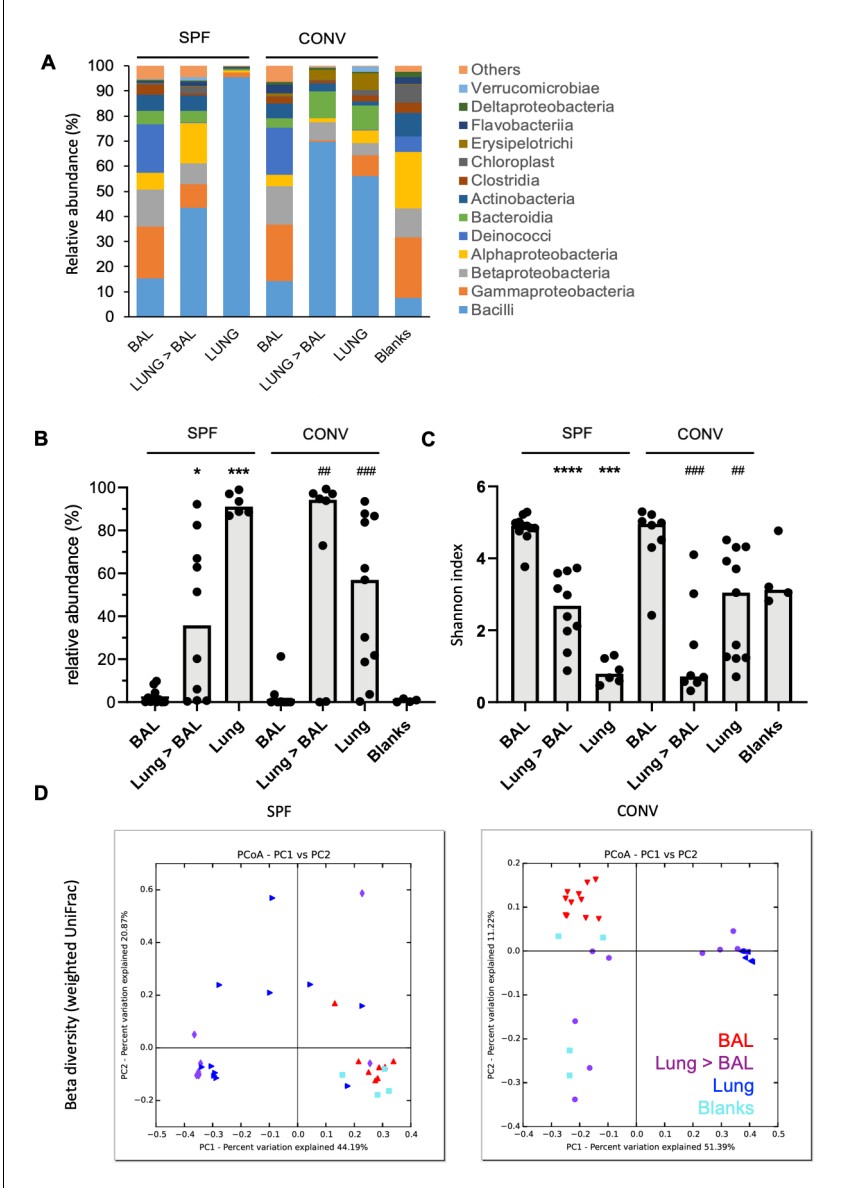

**Figure 2.** Bacilli class of bacteria is under-represented in bronchoalveloar lavage samples in mice in comparison to total lung homogenates. BALF, total lung homogenates after BAL procedure (Lung >BAL) or directly sampled total lung homogenates (Lung) of mice housed in specific pathogen-free environment (SPF) or housed conventionally (CONV) were prepped for bacterial 16S rRNA DNA libraries. Samples from mice housed in two different cages from two independent experiments for each group were pooled for analysis (SPF; $n_{BAL}$: 11, $n_{LUNG>BAL}$:10, $n_{LUNG}$:6, CONV; $n_{BAL}$: 8, $n_{LUNG>BAL}$:8, $n_{LUNG}$:11). Empty tubes were processed in parallel (Blank) for evaluation of contamination. (A) Mean relative abundance (%) of different taxonomical classes of bacteria in BAL, Lung >BAL, Lung and Blank samples. Color codes for each bacterial class are given next to the graph. Number of animals used for evaluation is indicated on the graph for each group (n). (B) Relative abundance (%) of Lactobacillus genus in BAL, Lung >BAL, Lung and Blank samples. Each circle represents an individual mouse. Medians of each group are depicted as gray columns. Mann-Whitney test is applied for statistical analysis (*: in comparison to BAL of SPF mice, #: in comparison to BAL of CONV mice). (C) Individual Shannon indices of BAL, Lung >BAL, Lungs, and Blank samples. Each black circle represents an individual mouse. Medians of each group are depicted as columns. Mann-Whitney test is applied for statistical analysis (*: in comparison to BAL of SPF mice, #: in comparison to BAL of CONV mice). (D) 2D PCoA plots (UniFrac, weighted) of BAL (red head-down triangles), Lung >BAL (purple circles), Lungs (blue head-left triangles), and blanks (turquoise squares).

The online version of this article includes the following figure supplement(s) for figure 2:

**Figure supplement 1.** Sensitivity controls for qPCR and 16S rRNA NGS.

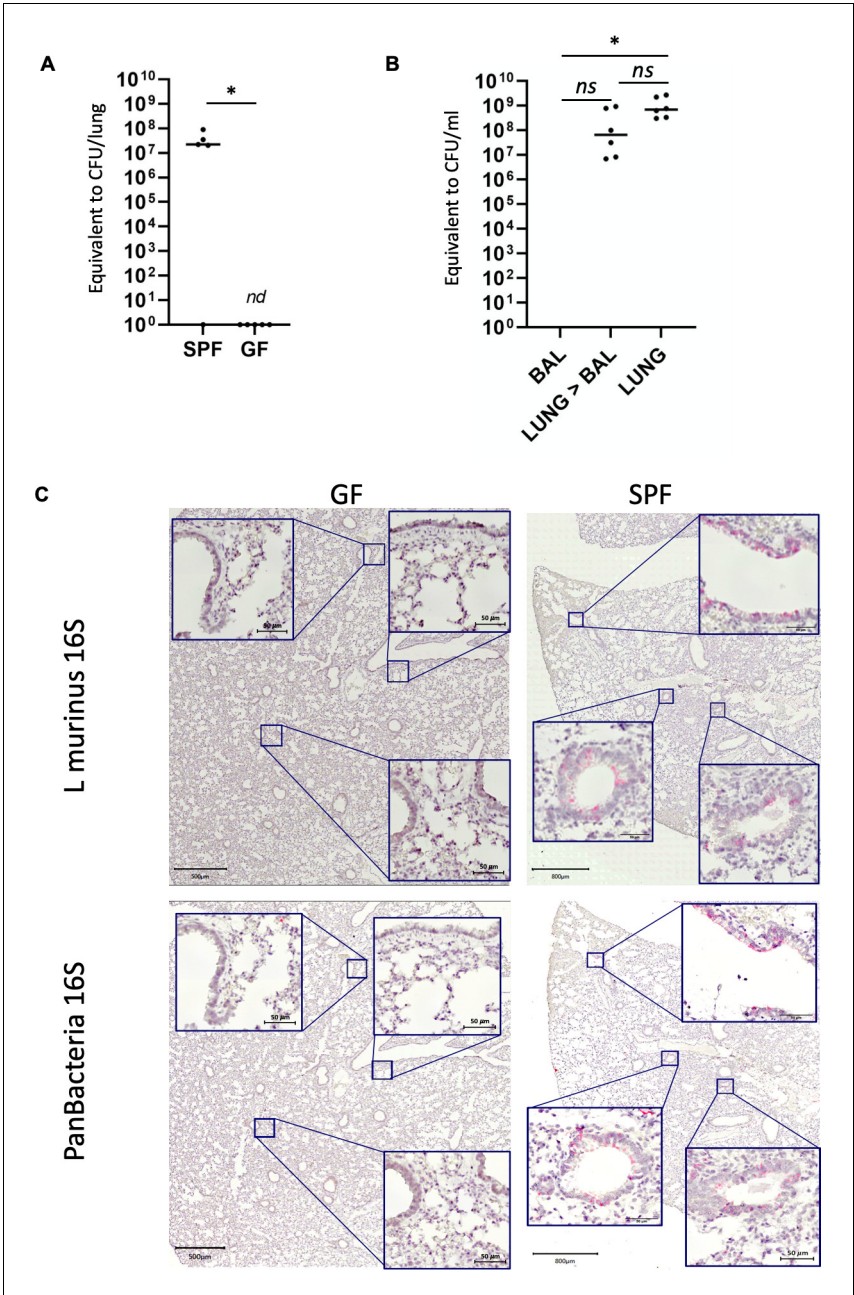

**Figure 3.** *L. murinus* is tightly associated with the respiratory tissue. (**A**) *L. murinus*-specific qPCR was used to determine genome copies of *L. murinus* in total lung DNA extracted from SPF mice (n = 5) or germ-free (GF) mice (n = 5). CT values were compared to standard of L. murinus genomes (see *Figure 3—figure supplement 3 and 1a*). 1/5 SPF mice and all GF mice revealed no specific amplicon (see also *Figure 3—figure supplement 3 and 1b*). Median value is indicated, each dot represents an individual mouse. (**B**) *L. murinus*-specific qPCR was used to determine genome copies of *L. murinus* in BALF (n = 11), Lungs after BAL (n = 10) and total lung (n = 6) DNA extracted from SPF mice. CT values were compared to standard of *L. murinus* genomes, median values are indicated (see *Figure 3—figure supplement 3 and 1a*). Kruskal–Wallis test was used to determine statistical significance in multiple comparisons with Dunn's correction. (**C**) Tissue distribution of L. murinus (upper panels) or total bacteria (lower panels) was determined in consecutive tissue slides of SPF mouse lungs (left panels) or GF mouse lungs (right panels). Bacterial colonization is indicated by a bright red/pink staining. Slides were counter-stained with hematoxylin. Representative sections of n = 5 mice for each group are shown.

The online version of this article includes the following figure supplement(s) for figure 3:

**Figure supplement 1.** Quality controls for L. murinus specific qPCR.

bacteria, *E. coli* (*Ec*CM). This *E. coli* strain was previously isolated from the LRT of mice (*Yildiz et al., 2018*) and characterized by whole genome sequencing. Phylogenetic analysis revealed close relationship with *E. coli* SJ7 (*Figure 4—figure supplement 1* and PRJNA591640 and *Supplementary file 2*). Only *Lm*CM inhibited growth of *S. pneumoniae* cultures that prevented reaching the same optical density, even after prolonged incubation, compared *S. pneumoniae* grown in FM or *Ec*CM supplemented TSB (*Figure 4A*). This result could be explained by the production of a soluble killing factor by *L. murinus* acting on *S. pneumoniae* and/or the inhibition of *S. pneumoniae* growth and division, without affecting viability. To decipher between the two possibilities, we conducted efficiency of plating assays of serially diluted *S. pneumoniae* cultures grown for 6 hr in FM, *Lm*CM, or *Ec*CM and found a one log in magnitude reduction of viability with *Lm*CM compared to the other conditions, confirming the reduction in bacterial growth (*Figure 4B,C*). This result suggests a soluble killing activity produced by *L. murinus* that acts on *S. pneumoniae*.

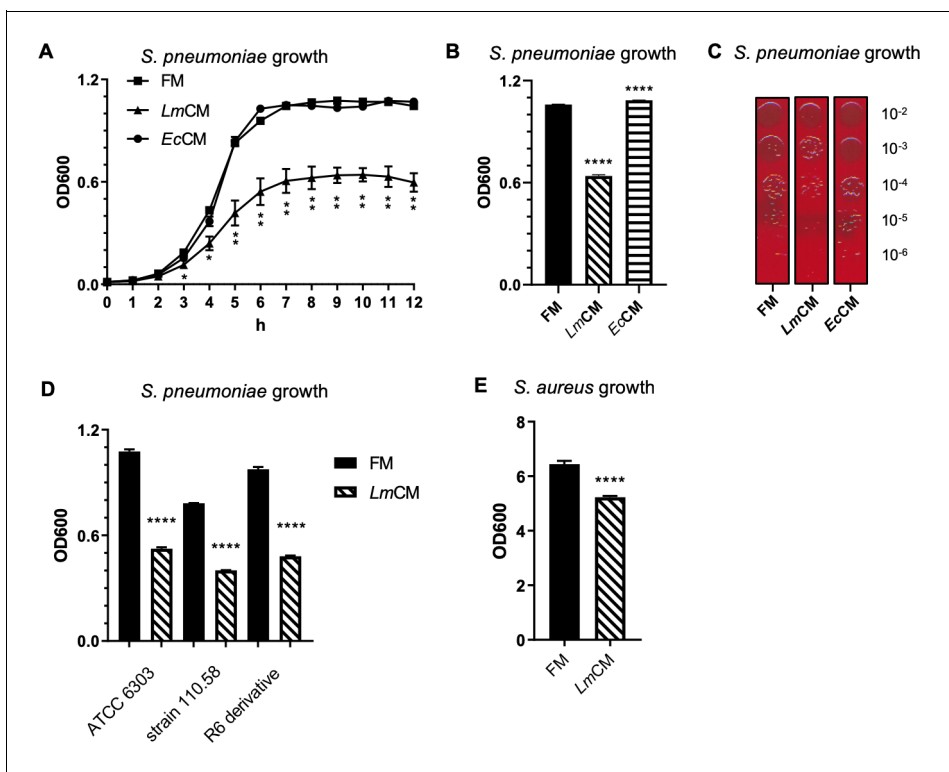

**Figure 4.** *L. murinus* conditioned media shows inhibitory role against *S. pneumoniae* growth in vitro. (A) *S. pneumoniae* cultures were grown in presence of fresh (FM) or *L. murinus* (*Lm*CM) or *E. coli* (*Ec*CM) conditioned media. Culture growth was followed hourly by optical density at 600 nm (OD600) for 12 hr. Pooled data from two independent experiments are given as mean ± SD. Student t-test was applied for significance test. (B) *S. pneumonia* cultures were grown for 6 hr in presence of FM, *Lm*CM, or *Ec*CM. Culture growth was measured by OD600. Representative data from two independent experiments are depicted as mean ± sd (n = 3). Student t-test was applied for significance test in comparison to fresh media treated group.( C) Following incubation in presence of FM, *Lm*CM, or *Ec*CM for 6 hr (note, the same OD was taken for the efficiency of plating at the 6 hr time point), *S. pneumonia* cultures are serially diluted and growth on fresh TSB agar plate with 5% sheep blood for 24 hr. Representative images are shown. (D) Different *S. pneumonia* strains, that is the virulent encapsulated strain (ATCC6303), non-encapsulated clinical isolate (110.58) and a derivate of the non-encapsulated avirulent lab strain R6 (R-6 derivative), or (E) *Staphylococcus aureus* (USA300) cultures were grown in presence of FM, *Lm*CM, or *Ec*CM for 6 hr. Culture growth was measured by OD600. Representative data from two independent experiments are depicted as mean ± SD. Student t-test was applied for significance test in comparison to fresh media group of each bacterial strain.

The online version of this article includes the following figure supplement(s) for figure 4:

**Figure supplement 1.** Phylogenetic tree of *E. coli* isolate to regular lab strains and other environmental isolates.

Next, we asked whether growth inhibition by *Lm*CM exhibits strain-specific differences on *S. pneumoniae*. To this end, we tested the activity of *Lm*CM on two additional strains of *S. pneumoniae*, i.e. the non-encapsulated clinical isolate (110.58) and a derivate of the non-encapsulated avirulent lab strain R6. Our results indicate that inhibition of *S. pneumoniae* by *Lm*CM is strain independent and is not affected by capsule formation since both strains displayed similar sensitivity to *Lm*CM as the virulent encapsulated strain (ATCC6303) (*Figure 4B–D*). Since a wide range of bacterial pathogens causes bacterial pneumonia, we wondered if the inhibition by *Lm*CM is specific to *S. pneumoniae*. *Staphylococcus aureus*, another Gram -positive bacterium, is a highly prevalence pathobiont, colonizing the upper respiratory tract of ~30% of the human population and frequently leading to bacterial pneumonia (*Wertheim et al., 2005*). Despite the much faster in vitro growth rate, *Lm*CM reduced optical density of *S. aureus* cultures (*Figure 4E*), implicating a broader acting mechanism against at least two distinct pathogenic bacteria.

Encapsulated *S. pneumonia* forms linear chains in vitro and in vivo. Formation of chains was previously associated with tissue adhesion and pathogenicity in vivo (*Rodriguez et al., 2012*). We observed a significant decrease in median chain length of *S. pneumonia* when grown in the presence of *Lm*CM (*Figure 5A/B*). *Ec*CM, however, did not affect *S. pneumoniae* growth and/or chain length. In order to narrow down the nature of the soluble inhibitor of *S. pneumoniae* growth, we first treated the *Lm*CM with Proteinase K. This treatment had no effect on its *S. pneumoniae* growth inhibition (*Figure 5—figure supplement 1*), excluding e.g. the action of bacteriocins. Our results suggested the active compound affecting *S. pneumoniae* growth in vitro could be a small bacterial metabolite.

## Secreted lactic acid is responsible for antibacterial effect of *L. murinus*

We first decided to identify metabolites specifically upregulated in *Lm*CM as compared to control (Fresh media, FM) or *Ec*CM with an untargeted multiplatform high-resolution mass spectrometry (HRMS) approach. 174 metabolite features that are at least 10-fold enriched in *Lm*CM over in FM and not upregulated in *Ec*CM (*Figure 5—figure supplement 1*) were detected. Two of these peaks were unambiguously identified by comparison to reference chemical standards: D-Glucosamine-6-phosphate and uridine-diphosphate-glucose. However, exposure of *S. pneumoniae* to these metabolites in concentrations up to 10 mM did not affect bacterial growth (*Figure 5—figure supplement*

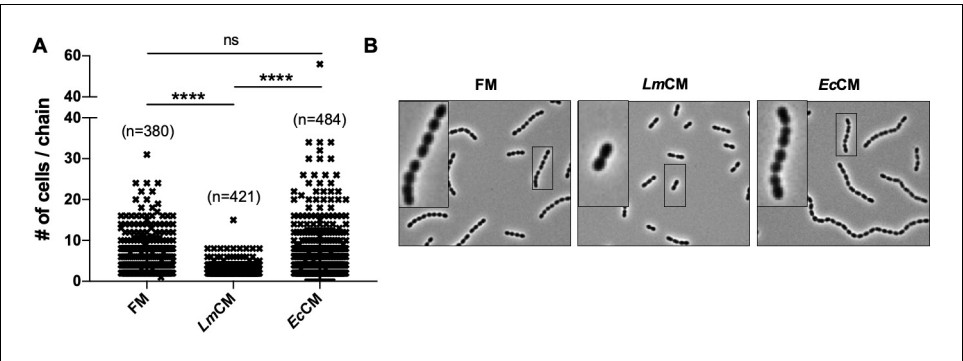

**Figure 5.** *Lm*CM limits cellular chain length of *S. pneumoniae* cultures. *S. pneumonia* cultures were grown to 0.6 OD600 in presence of FM, *Lm*CM or *Ec*CM. Cellular chain lengths of bacterial cultures are quantified using light microscopy. Each symbol represents an individual multicellular *S. pneumoniae* chain. Data was pooled from three independent experiments. Number of chains evaluated for each group indicated on the graph (**n**). Student t-test is applied for statistical analysis. Representative images are shown.

The online version of this article includes the following figure supplement(s) for figure 5:

**Figure supplement 1.** Active substance in *Lm*CM causing growth inhibition on *S. pneumoniae* is not protein in nature.

**Figure supplement 2.** Mass spectrometry identified hits from *Lm*CM do not cause similar growth inhibition on *S. pneumoniae* cultures.

**Figure supplement 3.** *Lm*CM dependent suppression of *S. pneumonia* growth does not rely on reactive oxygen species.

2), making it unlikely that any of these putative *L. murinus* metabolites are responsible for the observed inhibition of *S. pneumoniae* growth. The chosen HRMS approach would not allow us to detect very low molecular weight metabolites such as reactive oxygen species or lactic acid, which were both previously described as metabolites produced by Lactobacillaceae with potential anti-bacterial action (*Nardi et al., 2005*). Some Lactobacillus species were previously reported to produce high levels of $H_2O_2$ when grown in presence of oxygen (*Hertzberger et al., 2014*; *Kang et al., 2013*; *Marty-Teysset et al., 2000*). However, supernatants of *L. murinus* grown in hypoxia showed similar levels of growth inhibition (*Figure 5—figure supplement 3A*). Moreover, catalase treatment did not affect the growth arrest on *S. pneumoniae* seen with *Lm*CM, while it completely abolished the effect of $H_2O_2$ (*Figure 5—figure supplement 3B*). These results excluded reactive oxygen species from the candidate list of active metabolites in *Lm*CM.

Of note, *Lm*CM had a pH of 4.5 (*Figure 6A*). We observed a full recovery of *S. pneumoniae* growth, when the LCM was pH neutralized with NaOH (*Figure 6B*). We thus speculated that indeed

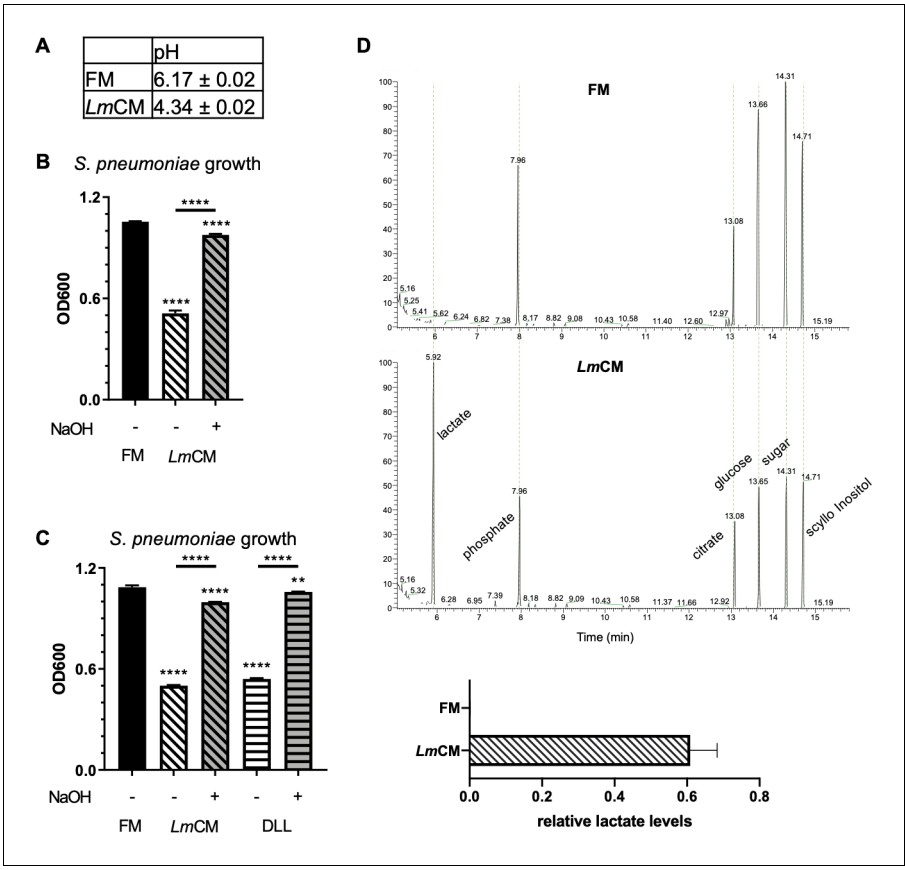

**Figure 6.** Lactic acid suppresses growth of *S. pneumoniae*. Lactic acid present in *Lm*CM is responsible for growth inhibition in *S. pneumoniae* cultures. (**A**) pH measurements of fresh (FM) or *L. murinus* conditioned (LmCM) media. Data is pooled from three independent experiments and depicted as mean ± SD (n = 3). (**B**) *S. pneumonia* cultures were grown for 6 hr in presence of FM, *Lm*CM or pH adjusted *Lm*CM (pH:~6.2, NaOH). Culture growth was measured by OD600. Representative data from two independent experiments are depicted as mean ± SD (n = 3). Student t-test was applied for significance test in comparison to fresh media treated group unless otherwise stated by the arrows. (**C**) Representative GC-MS total ion chromatograms of FM (dashed blue line) and *Lm*CM (straight red line) are shown. Concentrations of Lactate in each media, calculated based on a standard curve acquired using DL-Lactic acid, depicted in the graph (Mean ± SD, n = 3). Student t-test was applied for significance test. (**D**) S. pneumonia cultures were grown for 6 hr in presence of FM, LmCM, DL-Lactic acid (DLL, 100 mM) or pH adjusted LmCM (pH:~6.2, NaOH) and pH adjusted DLL (pH:~6.2, NaOH). Culture growth was measured by OD600. Representative data from two independent experiments are depicted as mean ± SD (n = 3). Student t-test was applied for significance test in comparison to fresh media treated group unless otherwise stated by the arrows.

lactic acid might be the compound responsible for the antimicrobial action of *Lm*CM. An involvement of lactic acid would be in accordance with the rescue of *S. pneumoniae* growth after pH neutralization we observed earlier (*Figure 6A/B*). Indeed, quantification of lactic acid in *Lm*CM by GC/MS revealed concentrations of approximately 100 µM (*Figure 6C*). In order to test if these concentrations are sufficient to explain the antibacterial effect of *Lm*CM, we grew *S. pneumoniae* for 6 hr in TSB complemented with 100 µM DL-lactic acid. Indeed, we observed inhibition of *S. pneumoniae* growth at comparable levels to *Lm*CM (*Figure 6D*), sustained over 24 hr (data not shown). These results suggest that lactic acid is the primary metabolite in *Lm*CM causing growth arrest on *S. pneumoniae*.

The second phenotype induced by LmCM, the reduction of median chain length of *S. pneumoniae* cultures (see *Figure 4a*), was not affected by presence of lactic acid (*Figure 7A*). Accordingly, pH neutralization did not restore chain length (*Figure 7B*). This implies that the effect of lactic acid on *S. pneumoniae* growth in vitro is independent of the previously observed inhibition of chain length. In order to characterize the nature of the compound responsible for inhibition of chain extension we repeated the treatment of *Lm*CM with protease K. Proteinase K completely abolished the reduced chain extension (*Figure 7C*), suggesting a secreted protein or peptide could be responsible for disrupting or preventing chain formation.

## Probiotic treatment with *L. murinus* reduces outgrowth of *S. pneumoniae* in vivo

In order to provide evidence that lung resident *L. murinus* does indeed provide colonization resistance against respiratory bacterial pathogens we inoculated germ-free (GF) mice and colonized SPF mice with either sterile PBS or 10e5 cfu *L. murinus* and challenged them 72 hr later with 10e3 *S. pneumoniae (ATCC 6303)*. 24 hr later we measured *S. pneumoniae* titers in total lung homogenates. To our great surprise we did not find increased susceptibility of GF mice to *S. pneumoniae* infection over colonized SPF mice (*Figure 8A*). In fact, GF mice were resistant to colonization, largely independent of *L. murinus* pretreatment. Resistance did not correlate with significantly lowered lung pH (*Figure 8B*). In colonized SPF mice, the pretreatment with *L. murinus* had a small but statistically non-significant effect on *S. pneumoniae* colonization (*Figure 8A*), which is likely due to the already present population of respiratory commensal *L. murinus*. Of note, *L. murinus* instillation did not alter body weight of the mice, which indicates no adverse effects by this probiotic treatment.

We thus decided to choose an alternative more realistic model, in which mice become more susceptible to *S. pneumoniae* colonization. Even in SPF mice, the colonization of the respiratory tract with low doses of *S. pneumoniae* is quite inefficient. This natural colonization threshold can be lowered by previous, sub-lethal IAV infection (*Figure 8B*), modeling the frequently observed secondary bacterial pneumonia in human IAV patients (*Morris et al., 2017*). In order to address if *L. murinus* could overcome the IAV mediated sensitization toward *S. pneumoniae* colonization, we established a respiratory probiotics treatment protocol (*Figure 8C*, probiotics treatment indicated by black arrow). When inoculating IAV infected mice on day 7-post infection (after onset of clinical signs of IAV infection) with *L. murinus* via the intranasal route, we observed a significant reduction BALF pH and diminished pneumococcal colonization on day 11-post IAV infection (24 hr post secondary *S. pneumoniae* challenge) (*Figure 8C/D*). Lactic acid levels in BALF of these mice were not changed compared to IAV infected and mock colonized controls (*Figure 8—figure supplement 1A*). We can currently not rule out that lactic acid is quickly taken up by the surrounding tissue or commensal bacteria. Importantly, treatment with *L. murinus* did not influence the overall pathology of IAV infection as indicated by similar weight loss in both experimental groups (*Figure 8—figure supplement 1C*). Lactobacilli were previously shown to modulate inflammatory signaling. An involvement of the innate host protein response was ruled out by specific qPCR for key cytokines involved in antibacterial signaling (*Figure 8—figure supplement 1B*). IAV infected mice showed a robust inflammatory response in their lungs on D10 post infection, which was non-significantly impacted by administration of *L. murinus*. Finally we tested by 16S rRNA gene-specific NGS, if *L. murinus* colonization of the lungs would alter the resident lung microbiota. Indeed, we found a mild increase in Bacilli in SPF mice colonized with L. murinus and a homogenous reduction of the remaining bacterial classes (*Figure 8—figure supplement 1C*). The overall composition of lung microbiota was however indistinguishable as indicated by largely overlapping clouds in the PCoA (*Figure 8—figure supplement*

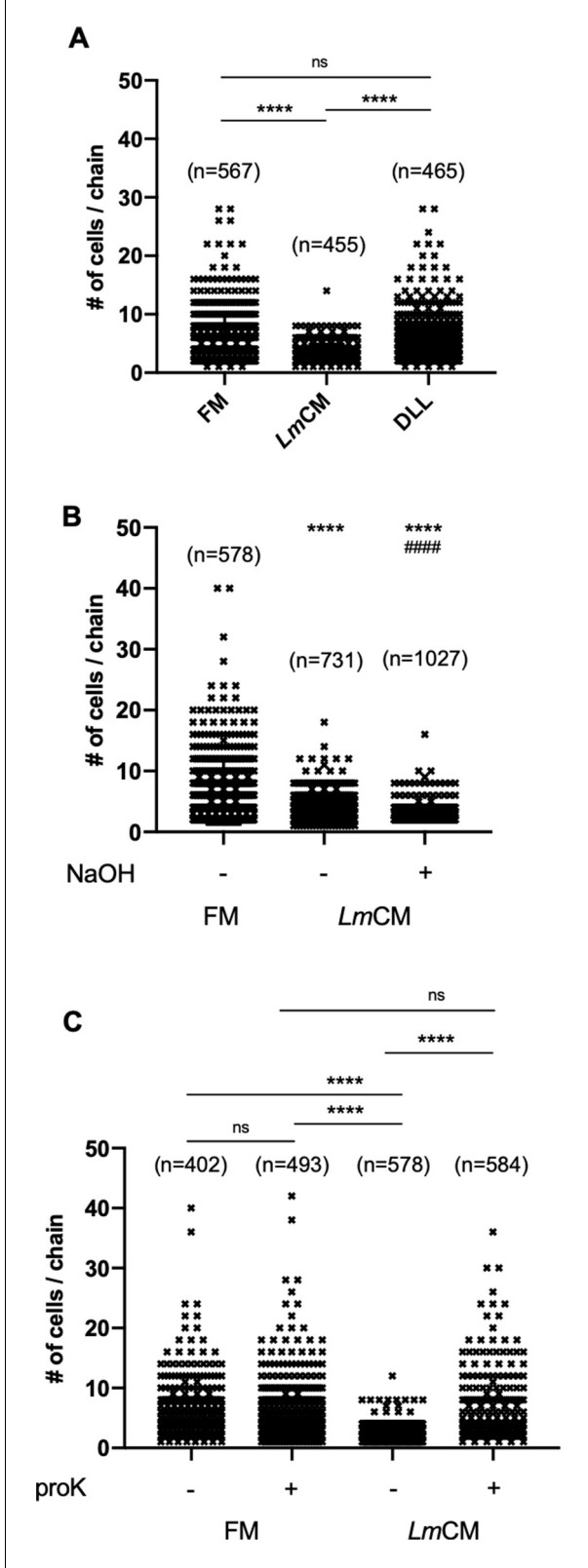

**Figure 7.** *L. murinus* secreted substance of protein origin is the cause of decrease in cellular chain length of *S. pneumoniae* cultures. *S. pneumonia* cultures were grown to 0.6 OD600 in presence of (**A**) FM, *Lm*CM or DL-Lactic acid (DLL,~100 mM), (**B**) FM, *Lm*CM or pH adjusted *Lm*CM (pH:~6.2, NaOH) and (**C**) mock or proteinase K (proK) treated FM and *Lm*CM. Cellular chain lengths of bacterial cultures are quantified using light microscopy. Each *Figure 7 continued on next page*

*Figure 7 continued*

symbol represents an individual multicellular *S. pneumoniae* chain. Data was pooled from three independent experiments. Number of chains evaluated for each group indicated on the graph (n). Student t-test is applied for statistical analysis.

*1D*). These data indicate that L. *murinus* could be a safe and directly acting antibacterial probiotic candidate for the respiratory tract.

## Discussion

The benefit of bacterial colonization of mucosal surfaces has been the center of attention of numerous scientific studies in recent years, ranging from nutritional science, to immunology and even behavioral studies. Most studies focus on the highly colonized digestive tract, where exposure of various immune cell types results in peripheral education of adaptive immune cells and fine-tuning of immune responses. Beyond these indirect effects, bacterial microbiota is posing a direct competitive threshold to colonization or outgrowth of pathogenic bacteria on barrier tissues.

Gnotobiotic animals, with an increased resistance to bacterial colonization, were first introduced in the mid-1960s by Russel W. Schaedler. This original Schaedler flora was later revised and became the ASF. Since the 1980s, all major commercial vendors provided mice for barrier facilities by colonizing germ-free animals with this standardized mix of eight bacterial strains. There are however numerous reports on vendor-specific variations in the commensal microbiota (*Ericsson et al., 2015*; *Guo et al., 2019*; *Ivanov et al., 2009*; *Ivanov et al., 2008*). Nevertheless, we believe our results are of importance for researchers working on lung physiology or pathology in commercially acquired mouse models.

In laboratory mice, *L. murinus*, as part of the ASF, was reported to colonize the intestine (*Almirón et al., 2013*; *Hemme et al., 1980*), the oral cavity (*Blais and Lavoie, 1990*; *Rodrigue et al., 1993*) and the vaginal tract (*Jerse et al., 2002*). In the lower respiratory tract *L. murinus* was to our best knowledge not reported as a dominant species, however Lactobacilli were previously found in high number in lung tissue microbiome (*Singh et al., 2017*; *Zhang et al., 2018*). Our data implicate that the sampling technique is critical for the adequate quantification of this species in the lung, since they might have a tight association with the host tissue. A previous comparison of BALF (+ or - cells) with lung tissue (from the distal tips of the lung) revealed enrichment of Firmicutes in tissue (*Barfod et al., 2013*), independently confirming our observation. Even within BAL samples of human patients, it was previously reported that a substantial proportion of commensal bacteria is tightly associated with cells and thus easily lost during removal of these cells (*Dickson et al., 2014*). From conventionally housed mice in our own facility, we know that Lactobacilli are still overall dominant in the lung tissue microbiome after two weeks of open cage housing.

Lactobacilli display a number of antimicrobial and immune modulatory activities. Depending on the ecological niche and the Lactobacillus strain investigated, diverse phenotypes have been observed: Colonization of GF mice with Lactobacillus spp. was proposed to increased alveoli numbers per $\mu m^2$ and to enhanced mucus production after 16 weeks (*Yun et al., 2014*), comparable to non-SPF mice, albeit this was not independently confirmed by others. Intranasal application of *L. rhamnosus* GG or *L. paracasei* one reduces IL-5 and eotaxin production in the lung (*Pellaton et al., 2012*). Intranasal application of *L. rhamnosus* GG was recently shown to suppress allergic responses in mouse models (*Spacova et al., 2019*). Colonization with *L. murinus* H12 was recently described to protect neonatal rats from necrotizing enterocolitis (*Isani et al., 2018*).

Direct antibacterial effects were also previously reported. Lactobacilli produce antimicrobial peptides such as bacteriocins (*Elayaraja et al., 2014*) and metabolites that are antibacterial in nature such as reactive oxygen species and lactic acid (*Nardi et al., 2005*). Our data imply a direct function for lactic acid in the antagonism of pneumococcus growth. Lactic acid was shown to inhibit growth of various bacterial pathogens; *E. coli*, *S. enterica* Typhimurium serovar, *L. monocytogenes* (*De Keersmaecker et al., 2006*; *Wang et al., 2015*). This action is assigned to two mechanisms: (1) its capacity to penetrate cytoplasmic membranes in its non-dissociated form, which results in lowered cytoplasmic pH and disruption of the proton gradient across the cytoplasmic membrane (*Alakomi et al., 2000*); 2) its capacity to permeabilize the cytoplasmic membrane, which was

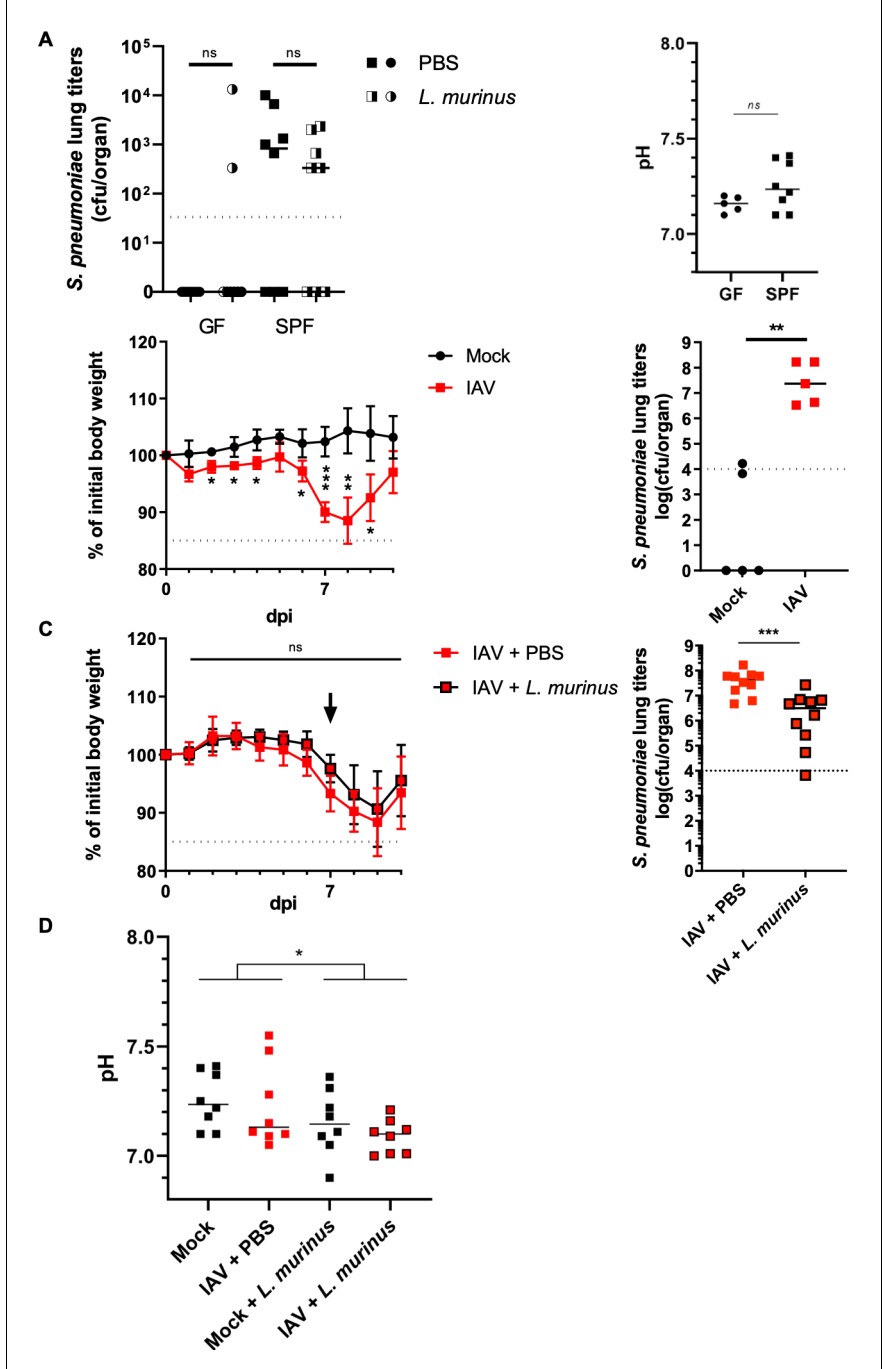

**Figure 8.** Therapeutic administration of *L. murinus* decreases *S. pneumoniae* titers in the lung of IAV infected mice. (**A**) Germ-free (GF) or specific pathogen-free (SPF) mice were intranasally administered with PBS or *L. murinus* (~$10^5$ cfu/animal). Three days post-administration, mice were challenged with *S. pneumoniae* (~1500 cfu/animal). *S. pneumoniae* lung titers (cfu/organ) of *L. murinus* (half black/half white) or PBS (plain black) administered GF (circles) or SPF (squares) are quantified 24 hr post infection (n = 10). Medians are depicted by a black line for each group. Pooled data from two independent experiments are shown. BALF pH does not significantly differ between colonized and GF mice (right panel). Median pH values for indicated mice are shown. Each dot represents one animal. Mann-Whitney test is applied for statistical analysis.( **B**) Average relative initial body weight (%)± SD is depicted for mock treated (black line with circles) and IAV infected (red line with squares) mice (n = 5). Mice were challenged with *S. pneumoniae* (~1500 cfu/animal) 10 days post IAV infection. *S. pneumoniae* lung titers (cfu/organ) of mock treated (black circles) or IAV infected (red squares) are quantified 24 hr post infection (n = 5). Medians are depicted by a black line for each group. Student t-test was applied for significance test. (**C**) All mice

*Figure 8 continued*

are infected with IAV (40 pfu). Seven days post IAV infection, PBS (red line with red squares) or *L. murinus* (~$10^5$ cfu/animal, black line with red squares) is intranasally administered to the animals. Average relative initial body weight (%)± SD is depicted for both group (n = 10). Mice were challenged with *S. pneumoniae* (~1500 cfu/animal) 10 days post IAV challenge. *S. pneumoniae* lung titers (cfu/organ) of IAV infected, PBS (red squares) or *L. murinus* administered (black rounded red squares) are quantified 24 hr post infection (n = 10). Medians are depicted by a black line for each group. Pooled data from two independent experiments are shown. Student t-test was applied for significance test. (D) pH of BALF from mock treated or *L. murinus* colonized mice 3 days post colonization and 10 days post IAV vaccination (n = 7–8 animals/group).

The online version of this article includes the following figure supplement(s) for figure 8:

**Figure supplement 1.** Probiotic treatment with L. murinus does not alter lactic acid levels, inflammatory response or composition of commensal lung bacteria.

proposed to increase in turn the sensitivity of lactic acid exposed bacteria (mainly gram-negative) to other antimicrobial compounds (*Alakomi et al., 2000*). Lactic acid levels where however not changed in vivo after exogenous application of *L. murinus*. We can thus not rule out that lactate independent mechanisms, e.g. the in vitro observed reduction of *S. pneumoniae* chain length, come into play in vivo. Additionally, we found a protease sensitive reduction of chain extension. It is currently unclear if both phenotypes contribute to the in vivo colonization defect of *S. pneumoniae* after probiotics treatment with *L. murinus*. The antibacterial activity was not limited to *S. pneumoniae* but also affected *S. aureus*. Published reports indicate a broad antibacterial activity of *L. murinus* (*Bilkova et al., 2011*; *Perelmuter et al., 2008*). In our study the repressing effects on fast growing *S. aureus* were mild but evident. The antimicrobial effect of *L. murinus* on *S. aureus* growth depends on low pH as well (*Bilkova et al., 2011*).

Notably we found that germ-free mice were more difficult to colonize with *S. pneumoniae*. This appears as a contradiction to the hypothesis that lung microbiota poses a colonization barrier toward invading bacterial pathogens. An explanation to this dilemma could be that S. pneumoniae colonization requires yet unknown anatomical conditions induced by commensal lung microbiota. Future tests in antibiotic treated mice might resolve this phenomenon.

The majority of human lung microbiota studies rely on aspirates or endoscopic brushes. In a recent tissue-biopsy based human lung microbiota study (*Yu et al., 2016*), Firmicutes were found in all regions of the respiratory tract, albeit not to the same extend as shown here in SPF mice. We can moreover not rule out that other bacteria might produce lactic acid in the lung of humans to prevent colonization. One of the most prevalent KEGG pathways retrieved from the human tissue microbiome was carbohydrate metabolism (*Yu et al., 2016*), which includes the pathway involved in lactic acid generation. Under chronic inflammatory conditions as found in IPF patients, lactic acid levels in lungs are elevated albeit the source of this lactate could not be determined (*Kottmann et al., 2012*). In cystic fibrosis patients, neutrophils were identified as a source for lactic acid in sputum (*Bensel et al., 2011*). The contribution of commensal respiratory tract bacteria and host cells to antibacterial lactate production in healthy humans remains to be addressed.

Finally, we demonstrated here that lung resident commensals might be applicable as probiotics to counteract lung colonization by pathogenic bacteria, without major effects on lung commensals. In contrast to other Lactobacillus species (*Ding et al., 2017*) *L. murinus* is not immunogenic when administered orally in mice (*Östberg et al., 2018*). This was also confirmed here by overall similar levels of cytokine mRNAs involved in antibacterial signaling. However, it remains currently unclear through which mechanism IAV might weaken the commensal shield against colonization, or if the here demonstrated effect of probiotics treatment is solely based on alterations of the host immune response. Nevertheless, in the context of a lung environment prone to bacterial colonization, e.g. as we induced by IAV infection, this treatment could reduce the risk of superinfection or improve the clinical outcome. This is of special interest in elderly human patients with high susceptibility to respiratory tract infections and reduced lung microbiome complexity (*Kovacs et al., 2017*).

## Conclusions

Lung tissue-associated microbiota of mice differ substantially from BALF microbiota, in containing large proportions of *L. murinus*. Future studies on respiratory microbiota should take this into

account. Under dysbiotic conditions, intranasal instillation of *L. murinus* decreases the burden of *S. pneumoniae* colonization. In light of the ongoing antibiotic resistance crisis, probiotic treatment of bacterial lung infection might offer therapeutic alternatives.

## Materials and methods

### Animal experiments

Power analysis (G*power 3.1) was used to estimate group size for mouse experiments was performed based on experimental variation obtained in a previous study aiming for a power of 0.95 (*Yildiz et al., 2018*).

C57BL/6J mice (female, 7–8 weeks of age) were purchased from Charles River Laboratories (France) and housed under SPF/BSL2 conditions or under conventional non-SPF conditions with open cages. All animals were housed for 7 days to adjust to housing conditions under a strict 12 hr light/dark cycle and fed ad libitum. Conventionally housed animals were kept in open cages 14 days before experiments. germ-free animals with C57BL/6J background were generated by the Clean Mouse Facility of Department of Biomedical Research of the University of Bern. They were born and raised in flexible film isolators in the at University of Bern, transferred aseptically into sterile IVC cages provided with sterile food and water and kept in the BSL2 unit of the animal facility of the Centre Medical Universitaire, Geneva for the duration of the experiments. For inoculation with IAV, *S. pneumonia* or *L. murinus*, mice were injected intraperitoneally with a mix of ketamine/xylazine (100 mg/kg and 5 mg/kg, respectively) in 200 μl of sterile PBS. Upon reaching deep anesthesia, mice were inoculated with 40 μl of PBS, virus or bacterial suspension via the intranasal route.

Body weights were measured daily during the light phase. Upon reaching experimental or humane endpoints (85% of initial body weight), animals were euthanized using controlled $CO_2$ exposure. Broncho alveolar lavages (BAL) or lungs were sampled immediately after euthanasia using sterile tools. BAL sampling was performed as previously described (*Sun et al., 2017*) with some modifications. Briefly, mice were sacrificed by controlled $CO_2$ inhalation and fixed with needles to a Styrofoam panel. The skin from abdomen to neck was cut open to expose thoracic cage and neck. With a second set of sterile tools soft tissue around the neck was gently removed. Using forceps, approximately 10 cm sterile sewing thread was placed underneath the trachea. With sterile scissors the trachea was nicked and a 22G × 1' Exel Safelet Catheter was inserted no more than 0.5 cm. The inner cannula was removed and the catheter was tied to trachea with a double knot in the sewing thread. 1 ml of PBS, or saline (0.9%) in experiments for pH or lactate measurements, was slowly introduced into the lung through the catheter with a 1 ml syringe, incubated for 10 s and aspirated back to the syringe. This procedure was repeated two more times, using a new sterile syringe in every step. Retrieved liquid was collected in a screw cap tube. Tools were changed in between organs and experimental groups to avoid cross contamination. BAL or lung samples were immediately stored at −80 °C until extraction of DNA.

### In situ hybridization

Lungs were collected under sterile conditions from five SPF and five germ-free C57BL/6J mice, fixed in paraformaldehyde 4% solution and embedded in paraffin and sliced at two different depth levels (51 μM and 101 μM), using a standard microtome (1 μM thickness).

In situ detection of *L. murinus* 16S rRNA was performed using the RNAscope 2.5 HD Assay - RED (Advanced Cell Diagnostics) according to the manufacturer's protocol. Briefly, lung slides were deparaffinized by submerging the slides four rounds in 100% xylol and subsequently four rounds in 96% ethanol solutions. Slides were dried at 60°C. An incubation with hydrogen peroxide was performed for 10 min and target retrieval was achieved by incubating the slides with RNAscope 1x Target Retrieval Reagent for 45 min at 95°C. Tissue was permeabilized using RNAscope protease plus (Advanced Cell Diagnostics). Probe hybridization was performed with a specific RNA probe for the 16S rRNA of *L. murinus* or 16S rRNA Panbacteria (#475131 or #451961 Advanced Cell Diagnostics) for 2 hr at 40°C. Signal amplification was achieved by incubating the tissues at 40°C with AMP solutions provided by the kit and a final Fast Red dye incubation for 10 min. Slides were counter-stained with hematoxylin and EcoMount solution was added to preserve the coloration. Slides were visualized under Olympus VS120 brightfield microscope (Zeiss) under a 100x/1.4 Oil objective and

processed using QuPath-0.2.1 Software. Random fields were picked from each slide and evaluated at 100x magnification. An average of 5–8 fields per sample were analyzed and positive staining was determined by red/pink punctate dots around epithelial cells of large airways.

## Bacteria

*L. murinus* and *Escherichia coli* isolates were grown from serially diluted lung homogenates on Columbia agar + 5% sheep blood plates (bioMerieux, France) at 37°C under aerobic conditions. *L. murinus* was cultured in vitro at 37°C with 5% $CO_2$ in static liquid broth or solid plates of MRS media (bioMerieux, France), prepared according to manufacturer's instructions. *E. coli* was grown in standard LB broth with agitation (250 rpm) or on LB plates, with antibiotic where indicated, at 37°C, except for conditioned media experiments where MRS media instead of LB were used in the same incubation settings. For experiments using *L. murinus* conditioned media (*Lm*CM) and *E. coli* conditioned media (*Ec*CM), overnight cultures were adjusted to 0.1 optical density at OD600 nm in MRS media. To grow *L. murinus* anaerobically, cultures were prepared as described above except in air-sealed plastic containers containing atmosphere generator, GenBox Anaer (bioMerieux, France). After 8 hr of incubation, bacteria were pelleted by centrifugation at 4000 x g at 4°C for 10 min. Supernatants were filtered through a 0.22 µm PVDF syringe-top filters (Merck Millipore, Ireland). Similarly processed fresh MRS media (FM) used as a negative control. *Streptococcus pneumoniae* strains, i.e. antigenic type 3 (ATCC-6303, LGC, Germany), Swiss non-encapsulated nasopharyngeal pneumococcal isolate 110.58 (MLST 344) and R6-derivative strain, were cultivated on Trypticase soy agar plates (bioMerieux, France) with 5% sheep blood (bioMerieux, France) at 37°C with 5% $CO_2$. Liquid cultures were grown directly from frozen stocks in Trypticase soy broth (bioMerieux, France) at 37°C with 5% $CO_2$ in static culture up to 0.5 optical density (OD600 nm). Then, cultures were diluted in Trypticase soy broth (bioMerieux, France) to 0.01 OD600, mixed with FM, *Lm*CM, or *Ec*CM in 1:10 ratio and incubated for 6 hr until experimental end point unless otherwise stated. Hydrogen peroxide ($H_2O_2$), UDP-glucose, glucosamine 6 P and DL-Lactic acid were purchased from Sigma-Aldrich (Switzerland). They were diluted in water or MRS media to desired concentration and used in same experimental setting as described above. *S. aureus* USA300 strain was a kind gift from the lab of Dr. William Kelly, University of Geneva, Switzerland. *S. aureus* cultures were grown and used in experiments in Trypticase soy broth (bioMerieux, France) at 37°C with agitation (250 rpm). For administration of *L. murinus* to mice, overnight cultures were diluted in fresh media (1:200) incubated in conditions aforementioned for 4 hr. Then, bacteria were pelleted down, washed once with PBS and diluted to ~$10^5$ cfu/animal in PBS prior to administration. *S. pneumoniae* inoculation was performed as previously described (*Yildiz et al., 2018*).

## Microscopy

Following incubation with fresh media, conditioned media or chemicals where indicated, *S. pneumoniae* cultures at optical density of 0.6 (OD600 nm) were installed on thin layer 1% pure agarose pads mounted on microscope slides. Alpha Plan-Apochromatic x 100/1.46 Ph3 (UV) VIS-IR oil objective on an Axio Imager M2 microscope (Zeiss) was used to visualize bacteria. A total number of 24–30 images of the samples were taken in triplicates; and representative images are shown. Number of individual cells per pneumococcal chain were counted to quantify the chain length.

## Bacteria DNA extraction, library construction, and bioinformatics analysis

Total lung, from lower trachea downward, and BAL samples were collected as described above. DNA extraction was performed using QIAGEN Pathogen Cador Mini kit (USA) according to manufacturer's protocol with slight modifications. Briefly, whole organ samples were homogenized with ¼" stainless steel grinding balls (MPBio, USA) or a QIAGEN Pathogen Cador Mini kit (USA) according to manufacturer's protocol using matched blanks in 1 ml PBS, containing 15 µl Proteinase K supplied with the kit, using a Bead Blaster 24 (Benchmark Scientific, USA) with a speed setting of 6 m/s for 30' and 30' break, repeated 10 times. Then, together with BAL samples containing same amount of Proteinase K, all samples were incubated at 56°C for 30'. 200 µl of homogenates were used for further DNA extraction steps following the kits' guidelines with pre-treatment T3 and B1. Empty sample tubes that are underwent the whole respective extraction procedure are used to evaluate to

measure potential cross contamination during PCR. Libraries for bacterial composition analysis of total lungs and BALs were performed as previously described (*Yildiz et al., 2018*). With exception to *Figure 8—figure supplement 1* QIIME was used for bioinformatics analysis of the sequences generated through Illumina (USA) sequencing through the pipeline previously described (*Aronesty, 2013*; *Caporaso et al., 2010a*; *Caporaso et al., 2010b*; *Chao, 1984*; *Colwell et al., 2012*; *DeSantis et al., 2006*; *Lozupone and Knight, 2005*; *Shannon, 1984*; *Wang et al., 2007*; *Yildiz et al., 2018*). For *Figure 8—figure supplement 1* we used QIIME2 for bioinformatics analysis of the sequences generated through Illumina (USA) sequencing through the following pipeline (*Bolyen et al., 2019*). Samples were demultiplexed and trimmed to remove adapters, primers and primer links using *cutadapt*. Subsequently, pair ends were joined using *vsearch* with the default parameters and denoised using *deblur* with a 250nt length limit. A table was generated containing the features for each sample and was further filtered to keep samples containing features with a frequency higher than 1140 using *feature-table*.

Taxonomy attribution for each feature was performed using a Naive Bayes classifier trained on the Greengenes_13_5 database at a 97% identity level with the primer sequences of the V4 rRNA 16S region. A relative frequency table was generated and bar plots were constructed by taking the mean ceiling of each group's taxa attribution at the class level. Beta-diversity was determined with Principal of Coordinates analysis by the weighted UniFrac method using **diversity core-metrics-phylogenetic** with a level of depth set to 500 on a rooted phylogenetic tree constructed with *phylogeny align-to-tree-mafft-fasttree*. 2D plots were constructed in emperor with the two axis that could explain most of the variability.

For whole genome sequencing, DNA was extracted from colony smears of *L. murinus* and *E. coli* using MOBIO DNA extraction kit according to manufacturer's protocol. Bacterial genomic DNA was quantified with a Qubit fluorimeter (Life Technologies). The Nextera kit from Illumina (USA) was used for the library preparation with 50 ng of DNA as input. Library molarity and quality was assessed with the Qubit and TapeStation using a DNA High sensitivity chip (Agilent Technologies). Libraries were loaded on a HiSeq 4000 single-read Illumina flow cell. Single-end reads of 50 bases and barcodes strategy were obtained according to the Nextera XT kit (Illumina, USA), following the manufacturer's recommendations. Read quality was assessed with the Fastqc program (available at: http://www.bioinformatics.babraham.ac.uk/projects/fastqc/) and filtered using the FastqMcf program (Ea-utils; available at: http://code.google.com/p/ea-utils). Genome assembly was performed using the SPAdes39. The sequencing run produced on average a total of 15.5 million reads per sample, exhibiting very high theoretical coverage values (between 150 and 250 fold). The assembly produced very similar results for each pair of strains, resulting in an average genome size of 2152198 bp and 5205614 nucleotides for *L. murinus* and *E. coli*, respectively.

Assembled genomes were annotated using the RAST program (*Treangen et al., 2014*). Multi-locus sequence typing (MLST) analysis was performed using annotated genomes and submitted to the Center for Genomic Epidemiology database (http://cge.cbs.dtu.dk/services/MLST). The phylogenetic relationship of all isolates was investigated by genomic single-nucleotide polymorphism (SNP)–based analysis using as the reference genome in the Parsnp v1.0 program (*Aziz et al., 2008*). The BlastP analysis was used to investigate the presence of specific genes involved in the phenotype, evolution, and virulence of the isolates.

The MAFFT Web Server42 was used to perform the phylogenetic analysis of 16 s rRNA gene of public sequences of Lactobacillus spp. Phylogenetic tree topologies of nucleotide 16S rRNA gene was constructed with the neighbor-joining method using bootstrap (the accession numbers of all species are provided in the figure). The evolutionary distances were computed using default parameters.

## Determination of average nucleotide identity (ANI) and in Ssilico DNA–DNA hybridization (DDH)

ANI- and DDH-values were assessed in silico using online tools. The assembled genome of strain Lacto_Soner was uploaded in GGDC (http://ggdc.dsmz.de/ggdc.php#) with the recommended local alignment tool BLAST+ and compared with the closest genomes available in public databases identified in the different phylogenetic trees, to obtain DDH-values. The accession numbers of the strains included in the phylogenetic analyses are provided in *Supplementary file 1*. The statistic

comparison (logistic regression) used a significant probability value of DDH >79%. Pairwise ANI-values were obtained using JSpeciesWS (http://jspecies.ribohost.com/jspeciesws/#analyse) with BLAST.

## Phylogenetic analysis

The phylogeny presented is based on the alignment of approximately 1400 nucleotides of the 16S rRNA gene. The phylogenetic analyses were generated with the neighbor-joining method. The percentage of replicate trees in which the associated taxa clustered together in the bootstrap test (100 replicates) is shown next to the branches. The trees are not rooted and drawn to scale, with branch lengths in the same units as those of the evolutionary distances used to infer the phylogenetic tree. The evolutionary distances were computed using the Kimura 2-parameter method for 16S rRNA gene. The analysis included 17 sequences. Evolutionary analyses were conducted using MEGA6. All sequences are labeled according to strain name, species and accession number.

## *L. murinus*-specific qPCR

For 16S rRNA gene DNA quantification, 200 ng of DNA extracted from organ homogenates were mixed with 10 µl of 2X ddPCR supermix for probes (BioRad), 1.8 µl of each Forward (10 µM) and Reverse (10 µM) primer, 0.4 µl of a DNA fluorescent probe and completed the volume to 20 µl with RNAse, DNase Free Molecular Biology Grade Water (Amimed, BioConcept, Switzerland). Quantitative PCR was performed following a thermal cycling protocol of an initial denaturation step at 95°C for 5 min, followed by 80 cycles of denaturation at 95°C for 30 s and annealing/extension at 61.5°C for 60 s. *L. murinus* 16S rRNA gene-specific primers and probe were used (probe: 5'-FAM-CTCAACCGTGCCGTTCAAACTG-MQ530-3'; 16Sfw: 5'- ACTGGCGATGTTACCTTTGG −3' and 16Srev: 5'- CAGGCCTTTGTATTGGTGGT −3'). Amplicons were loaded on an 2% agarose gel stained with ethidium bromide and visualized with a BioRad Gel DocTM XR+ (BioRad).

For the standard curve construction and colony-forming unit (CFU) correlation, serial dilutions of *L. murinus* genomic DNA extracted from a colony grown overnight as previously described were added to the mix as described above with the same thermal cycling protocol settings. Briefly, 2 ml of *L. murinus* grown in MRS broth were divided in equal volumes and: (1) washed in PBS twice to remove remaining broth, heated at 100°C for 5 min and incubated in distilled water for 2 min. (2), serial-diluted and plated on MRS agar overnight to determine CFU. Genomes/lung calculation was adjusted by correcting to the amount of total DNA extracted from each lung and the total DNA input on each qPCR.

For RNA levels where stated, 500 ng total RNA was used to synthesize cDNA using M-MLV Reverse Transcriptase kit following kit's guidelines (PROMEGA). Quantitative PCR was performed using 2X KAPA SYBR FAST qPCR Master Mix-universal according to manufacturer's instructions. The following specific primers for murine cDNAs were used: mIP-10 fw: 5'-TTCACCATGTGCCATGCC-3', mIP-10 rev: 5'-GAACTGACGAGCCTGAGCTAGG-3', mIFN-γ fw: 5'-ATGAACGCTACACACTGCATC-3', mIFN-γ rev: 5'-CCATCCTTTTGCCAGTTCCTC-3', mTNF-α fw: 5'-AGAAACACAAGATGCTGGGA-CAGT-3', mTNF-α rev: 5'-CCTTTGCAGAACTCAGGAATGG-3', 18S fw: 5'-GTAACCCGTTGAACCCCATT-3', 18S rev: 5'-CCATCCAATCGGTAGTAGCG-3'. Gene expression was normalized to 18S and fold induction was calculated compared to the mock group using the ΔΔCq formula.

## Identification of metabolites in *Lm*CM and *Ec*CM

200 µl of fresh media, *Lm*CM or *Ec*CM were de-proteinized by adding 800 µl of ice-cold MeOH. After vortexing for 1 min, samples were centrifuged at 15000 xg and 4°C for 15 min. Supernatants were collected and evaporated under vacuum. Residues were re-dissolved in 500 µl of MeCN:$H_2O$ (50:50, v:v) and vortexed for 1 min. Samples were centrifugated under the same conditions prior to injection in the LC-MS system. Chromatography was performed on a Waters H-Class Acquity UPLC system composed of a quaternary pump, a column manager and an FTN autosampler (Waters Corporation, Milford, MA, USA). For RPLC analyses, samples were separated on a Kinetex C18 column (150 × 2.1 mm, 1.7 µm) and the corresponding SecurityGuard Ultra precolumn and holder (Phenomenex, Torrance, USA). Solvent A was H2O and solvent B was MeCN, both containing 0.1% formic acid. The column temperature and flow rate were set at 30°C and 300 µl min$^{-1}$, respectively. The gradient elution was as follows: 2 to 100% B in 14 min, hold for 3 min, then back to 2% B in 0.1 min and re-equilibration of the column for 7.9 min. aHILIC separations were conducted on a Waters Acquity

BEH Amide column (150 × 2.1 mm, 1.7 µm) bearing an adequate VanGuard pre-column. Solvent A was H2O:MeCN (5:95, v/v) and solvent B was H2O:MeCN (70:30, v/v) containing 10 mM ammonium formate (pH = 6.5 in the aqueous component). The following gradient was applied: 0% B for 2 min, increased to 70% B over 18 min, held for 3 min, and then returned to 0% B in 1 min and to re-equilibrate the column for 7 min (total run time was 31 min). The flow rate was 500 µl min–1, and the column temperature was kept at 40°C. For the zHILIC method, separation was performed on a Merck SeQuant Zic-pHILIC column (150 × 2.1 mm, 5 µm) and the appropriate guard kit. The following gradient of mobile phase A (MeCN) and mobile phase B (2.8 mM ammonium formate adjusted to pH 9.00) was applied: 5% B for 1 min, increased to 51% B over 9 min, held for 3 min at 51% B and then returned to 5% B for 0.1 min before re-equilibrating the column for 6.9 min (total run time was 20 min) at a flow rate of 300 µl/min and a column temperature of 40°C.

In all cases, a sample volume of 0.5 µl was injected. Samples were randomized for injection, and QC pools were analyzed every six samples to monitor the performance of the analytical platform.

The UPLC system was coupled to a maXis 3G Q-TOF high-resolution mass spectrometer from Bruker (Bruker Daltonik GmbH, Bremen, Germany) through an electrospray interface (ESI). The instrument was operated in TOF mode (no fragmentation). The capillary voltage was set at −4.7 kV for ESI+, drying gas temperature was 225°C, drying gas flow rate was set at 5.50 (RPLC), 8.00 (aHILIC) or 7.00 (zHILIC) L min-1 and nebulizing gas pressure was 1.8 (RPLC) or 2.0 bar (HILIC). Transfer time was set at 40 (RPLC) or 60 (HILIC) µs and pre-pulse storage duration at 7.0 (RP) or 5.0 µs (HILIC). For ESI– operation, the capillary voltage was set at 2.8 kV. All the remaining ion source and ion optics parameters remained as in ESI+. Data between 50 and 1000 m/z were acquired in profile mode at a rate of 2 Hz. ESI and MS parameters were optimised using a mix of representative standards fed by a syringe pump and mixed with the LC eluent (mid-gradient conditions) within a tee-junction. Format adducts in the 90–1247 m/z range were employed for in-run automatic calibration using the quadratic plus high-precision calibration algorithm provided by the instrument's manufacturer. MS and UPLC control and data acquisition were performed through the HyStar v3.2 SR2 software (Bruker Daltonik) running the Waters Acquity UPLC v.1.5 plug-in.

Run alignment, peak piking and sample normalization were performed on Progenesis QI v2.3 (Nonlinear Dynamics, Waters, Newcastle upon Tyne, UK) and peaks were identified by matching their retention times, accurate masses and isotopic patterns to those of a library of chemical standards (MSMLS, Sigma-Aldrich, Buchs, Switzerland) analyzed under the same experimental conditions, as described elsewhere (*Pezzatti et al., 2019*).

## Measurement of lactic acid levels in *Lm*CM

D-/L-Lactic acid (D-/L-lactate) (Rapid) Assay Kit (Megazyme, USA) was used to quantify lactic acid levels in bacterial supernatants according to manufacturer's instructions. For GC-MS, samples were diluted 1:10 in ultra-pure water. 5 µl of diluted fresh or spent medium were transferred to GC-MS vial inserts (Machery-Nagel, 702813) and dried down in a centrifugal evaporator (Speed Vac Concentrator, Savant) together with 10 µl scyllo-inositol (1 mM, Merck, I8132) as internal standard. Samples were re-dried after addition of 20 µl methanol (Merck, 34860). Vial inserts were transferred to 1.5 ml vials suitable for mass spectrometry (Machery-Nagel, 702282) and subsequently derivatised through addition of 20 µl pyridine and 20 µl bis(trimethylsilyl)trifluoroacetamide +1% trimethylchlorosilane (BSTFA +1% TMCS, Merck, B-023).

Samples were analyzed by GC-MS after >30 min incubation at room temperature using a Trace GC Ultra (Thermo Fisher Scientific) gas chromatography system equipped with a Phenomenex ZB-5MS capillary column (30 m x 0.25 mm, 0.25 µm, with inert guard). Using a Triplus RSH autosampler (Thermo Fisher Scientific) 2 µl of sample were injected. The GC was operated in splitless mode for one minute followed by split mode (1/13). The helium carrier gas flow rate was set to 1 ml/min, the temperature of the injector and transfer line were set to 270°C and 320°C, respectively. The oven temperature was at 70°C (1 min hold), increased to 295°C (12.5 °C/min) and raised to 320°C (25 °C/min, 2 min hold). The GC was coupled to a PolarisQ ion trap mass spectrometer (Thermo Fisher Scientific), operated in electron ionization (EI) mode (70 eV). The ion source was operated at 200°C and full scan data (m/z 50–650) was acquired after a solvent delay of 5 min. Relative levels of lactate were quantified by determining the intensity of the lactate ion m/z 219 relative to an abundant ion of the internal standard (m/z 318) using XCalibur (Thermo Fisher Scientific), OpenChrom and Excel (Microsoft).

## pH measure in BALF

BALF were collected as described above. For pH measurement, samples were thawed on ice and the HALO – HANNA pH probe (Hanna Instruments) was submerged in each sample with a distilled water wash in between. pH was determined with the Edgeblu – HANNA pHmeter (Hanna Instruments) calibrated on the same day of the analysis.

## Virus

Reverse genetics systems were kindly provided by Dr. Peter Palese and Dr. Adolfo García-Sastre (Icahn School of Medicine at Mount Sinai, New York, NY, USA). IAV A/Viet Nam/1203/2004 (VN/1203) HALo (low pathogenic version) was rescued and stocks were prepared as previously described (*Anchisi et al., 2018*; *Eisfeld et al., 2014*).

## Statistics

In order to determine statistical significance, we applied Mann–Whitney test to OTU abundance, Shannon indices and bacterial titers from mouse organs using Graph Pad Prism 7.0. In vitro experiments were assessed with student's t-test. Where p is not indicated; ****: $p \leq 0.0001$, ***: $p \leq 0.001$, **: $p \leq 0.01$, *: $p \leq 0.05$, ns: not significant.

## Ethical approval

All animal procedures were in accordance with federal regulations of the Bundesamt für Lebensmittelsicherheit und Veterenärwesen (BLV) Switzerland (Tierschutzgesetz) and approved by an institutional review board and the cantonal authorities (license number GE-159–17).

## Consent for publication

All authors read the manuscript and agreed to its publication.

## Acknowledgements

We are grateful for excellent technical assistance by Chengyue Niu and Filomena Silva. We thank Ingrid Wagner from the team of Dr. Doron Merkler, University of Geneva, for providing guidance in setting up the RNAScope technique. We would like to extend our gratitude to the axenic mouse facility at the University of Bern and Dr. Mercedes Gomez, to the technical support of the mouse facility at the CMU, to the excellent support by the Genomics core facility, the Histology core facility and the Bioimaging facility of the CMU.

## Additional information

### Funding

| Funder | Grant reference number | Author |
|---|---|---|
| Swiss National Science Foundation | 155959 | Mirco Schmolke |
| Swiss National Science Foundation | 182475 | Mirco Schmolke |
| Fondation Ernst et Lucie Schmidheiny | | Mirco Schmolke |
| Fondation Ernest Boninchi | | Inês Boal-Carvalho |
| Swiss National Science Foundation | 162808 | Lucy J Hathaway |
| Swiss National Science Foundation | 169791 | Siegfried Hapfelmeier |
| Swiss National Science Foundation | 182576 | Patrick H Viollier |
| H2020 European Research Council | 695596 | Joachim Kloehn |

The funders had no role in study design, data collection and interpretation, or the decision to submit the work for publication.

## Author contributions

Soner Yildiz, Conceptualization, Data curation, Formal analysis, Investigation, Visualization, Methodology, Writing - original draft; João P Pereira Bonifacio Lopes, Inês Boal-Carvalho, Investigation; Matthieu Bergé, Joachim Kloehn, Formal analysis, Investigation, Methodology; Víctor González-Ruiz, Resources, Formal analysis, Investigation, Methodology; Damian Baud, Patrice Francois, Data curation, Formal analysis, Investigation; Olivier P Schaeren, Investigation, Methodology; Michael Schotsaert, Resources; Lucy J Hathaway, Methodology; Serge Rudaz, Supervision, Validation, Investigation, Methodology; Patrick H Viollier, Supervision, Funding acquisition, Writing - original draft; Siegfried Hapfelmeier, Supervision, Investigation, Methodology; Mirco Schmolke, Conceptualization, Supervision, Funding acquisition, Writing - original draft, Project administration

## Author ORCIDs

Matthieu Bergé (iD) http://orcid.org/0000-0002-0910-6114
Patrick H Viollier (iD) http://orcid.org/0000-0002-5249-9910
Mirco Schmolke (iD) https://orcid.org/0000-0002-2491-3029

## Ethics

Animal experimentation: All animal procedures were in accordance with federal regulations of the Bundesamt für Lebensmittelsicherheit und Veterenärwesen (BLV) Switzerland (Tierschutzgesetz) and approved by an institutional review board and the cantonal authorities (license number GE-159-17).

## Decision letter and Author response

Decision letter https://doi.org/10.7554/eLife.53581.sa1
Author response https://doi.org/10.7554/eLife.53581.sa2

# Additional files

## Supplementary files

• Supplementary file 1. Genome annotation of *L. murinus* isolate. *L. murimus* genome annotation obtained by RAST showing contig identifiers and detected features, start, stop coordinates and coding strand are indicated considering corresponding contig. Enzymatic commission number (EC), nucleotide and deduced amino acid sequences are also presented.

• Supplementary file 2. Analysis of pathogenicity islands in *E. coli* genome isolated from IAV infected mice. The tables contain the list of genes putatively present onto pathogenicity islands recovered from the annotated genomes of two individual colonies of *E. coli* (**A** and **B**) isolated from lung tissue. Gene name and short name as well as gene function are presented. Note that the two lists have almost exactly the same content.

• Supplementary file 3. Features found in metabolomics analyses of FM, LmCM or EcCM, named after their analytical technique, retention time, and mass. Sheet 1 (whole list) contains all hits sorted by the ratio of the average signal (arbitrary units) from two runs for L. murinus conditioned medium (LmCM) over fresh medium (FM). Sheet two lists the hits with an AVG (LmCM)/AVG (FM) >10 fold. Sheet three lists the hits with an AVG *E. coli* conditioned medium (EcCM)/AVG (LmCM) <10 fold. Sheet 4 (short list) contains only hits that are present in sheet 2 and 3. Sheet five lists identified metabolites based on their masses and retention times compared to authentic reference standards run in-house under the same analytical setup. Score (arbitrary units) indicates the goodness of the overall matching between the experimental properties of each feature and those obtained for the corresponding standard, as calculated by Progenesis QI. Mass error express the relative mass difference (ppm) between the masses of the feature and the reference standard. Isotope similarity (%) corresponds to the matching of the isotopic profile of the feature and the reference standard.

• Transparent reporting form

## Data availability

Material from this study originally generated in our team, will be made available upon request in reasonable quantities.

The following datasets were generated:

| Author(s) | Year | Dataset title | Dataset URL | Database and Identifier |
|---|---|---|---|---|
| Schmolke M | 2019 | NGS MiSeq data | http://www.ncbi.nlm.nih.gov/bioproject/591377 | NCBI BioProject, PRJNA591377 |
| Schmolke M | 2019 | Lactobacillus murinus isolated from laboratory mice | https://www.ncbi.nlm.nih.gov/bioproject/PRJNA663937 | NCBI BioProject, PRJNA663937 |
| Schmolke M | 2019 | *Escherichia coli* isolated from laboratory mice | https://www.ncbi.nlm.nih.gov/bioproject/PRJNA591640 | NCBI BioProject, PRJNA591640 |
| Schmolke M | 2020 | NGS MiSeq data | http://www.ncbi.nlm.nih.gov/bioproject/663937 | NCBI BioProject, PRJNA663937 |

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
