## [Decision Letter]

**Acceptance summary:**

The lung microbiome is an emerging area of study with clear clinical significance. While the mechanisms through which commensal bacteria in the gut protect against enteric pathogens has been well described, the role of lung commensals has not been clearly elucidated. This work identifies an enrichment of the commensal bacterium *Lactobacillusmurinus* in mouse lung tissue, which could potentially protect the host against pneumococcal colonization.

**Decision letter after peer review:**

Thank you for submitting your article "Respiratory tissue-associated commensal bacteria offer therapeutic potential against pneumococcal colonization" for consideration by *eLife*. Your article has been reviewed by Wendy Garrett as the Senior Editor, a Reviewing Editor, and three reviewers. The following individual involved in review of your submission has agreed to reveal their identity: Kenneth Klingenberg Barfod (Reviewer #3).

Summary:

Yildiz et al., describe the enrichment of *Lactobacillusmurinus* in mouse lung tissue, which could potentially protect the host against pneumococcal colonization. The lung microbiome is an emerging area of study with potential clinical significance. While the mechanisms through which commensal bacteria in the gut protect against enteric pathogens has been well described, the role of lung commensals has not been clearly elucidated. Here, they show that *L. murinus* suppresses the lung pathogen *S. pneumoniae*. SPF mice have high levels of L. murinus. Conditioned media suppresses *S. pneumoniae* growth likely due to lactic acid production. Finally, they show a significant reduction in *S. pneumoniae* levels in mouse models in the presence of L. murinus.

Essential revisions:

Addition of qPCR and/or FISH data: The authors did not show the 16S qPCR data to compare the overall bacterial load between BAL, total lung homogenate, and lung homogenate after BAL. However, such data is critical to demonstrate the difference of commensal bacteria that are reside in the airway vs. those are tissue-associated. While the authors mentioned the 16S copy is higher in tissue-derived samples than BALF samples, and the BALF samples are indistinguishable from background, it is necessary to directly show the results, with the proper normalization and control. In particular, adjacent sterile tissue (such as heart) could be used as a control for lung tissue samples. Alternatively, FISH could be performed to directly demonstrate the presence of *L. murinus* in the lower respiratory tract and their association with lung tissue.

In vivo measurements of pH and lactic acid levels: It is not clear if the proposed mechanism is physiologically relevant. Data on pH and lactic acid concentration of BALF samples from *L. murinus* colonized versus GF mice would be a good addition to the paper. This mechanism also seems to contradict the proteinase K experiment, which implicates a protein/peptide.

The authors should evaluate the alternative hypothesis that the impact of *L. murinus* is mediated through interactions with the immune system. As a first pass, the authors could perform cytokine analysis from the BAL/tissue of the animals or ideal repeat the experiment with flow cytometry.

The authors observe that the β diversity and 16S qPCR copy numbers of BAL and blanks (controls) were similar. This is surprising given that several other studies in mouse and humans (Charlson et al., 2012, Dickson, 2014) have clearly demonstrated significant differences in controls and BAL. The authors suggest that this indicates a high number of false positives. What do the authors think is the cause for this? Does it reflect an issue with contamination of the saline/sterile water used for BAL, their sequencing techniques, or something else?

Need to deposit sequencing data prior to resubmission. This includes the sequencing data from lung or bal of the mice from both model systems that were inoculated with L. murinus. This will provide insight into the other commensals or pathobionts that may have been are promoted or suppressed by *L. murinus*. It is possible that growth of other pathobionts/commensals in the lung due to *L. murinus* exposure may be in part or indirectly responsible for suppression of *S. pneumoniae*.

[Editors' note: further revisions were suggested prior to acceptance, as described below.]

Thank you for submitting your article "Respiratory tissue-associated commensal bacteria offer therapeutic potential against pneumococcal colonization" for consideration by *eLife*. Your article has been reviewed by Wendy Garrett as the Senior Editor, a Reviewing Editor, and three reviewers. The following individuals involved in review of your submission have agreed to reveal their identity: Nirmal Sharma (Reviewer #2); Kenneth Klingenberg Barfod (Reviewer #3).

The reviewers have discussed the reviews with one another and the Reviewing Editor has drafted this decision to help you prepare a revised submission.

Summary:

Yildiz et al., describe the enrichment of *Lactobacillusmurinus* in mouse lung tissue, which could potentially protect the host against pneumococcal colonization. The lung microbiome is an emerging area of study with potential clinical significance. While the mechanisms through which resident bacteria in the gut protect against enteric pathogens has been well described, the role of the lung microbiota has not been clearly elucidated. Herein, the data show that *L. murinus* suppresses the lung pathogen *S. pneumoniae*. L. murinus-conditioned media suppresses *S. pneumoniae* growth likely due to lactic acid production. Finally, the authors find a significant reduction in *S. pneumoniae* levels in mouse models in the presence of *L. murinus*.

Essential revisions:

The reviewers request that multiple claims in the paper be modified or weakened, setting the stage for future studies. The appropriate caveats could be added to the results or Discussion section. This includes:

It is still not clear what the overall bacterial load is in BAL, total lung homogenate, and lung homogenate after BAL by 16S qPCR. Without such data, there is not sufficient evidence to claim *L. murinus* is tightly associated with lung tissue.

Regarding the physiological relevance of lactate derived from L. murinus, the authors didn't observe any changes 3 days post *L. murinus* inoculation in mock or influenza A virus pre-treated mice.

Cytokine levels are interesting but would require flow cytometry and additional experiments to better understand how the immune system is involved.

Contamination introduced from the DNA preparation kit would also be present in lung samples.

Subsection “*Lactobacillus murinus* (ASF361) is a major constituent of mouse lung microbiota”: Please revise this subheading to be more specific to the current study, unless the authors meant to imply that *L. murinus* is a major player in the lung microbiota in all mice.

Please discuss why germ-free mice are resistant to *S. pneumoniae* infection regardless of *L. murinus* pre-treatment.

Make sure to include the appropriate statistical analysis and comparison groups in each figure.

Subsection “*Lactobacillus murinus* (ASF361) is a major constituent of mouse lung microbiota”: Show the data on 16S copy numbers based on qPCR.

Materials and methods section: It is imperative to describe how many cages, how they were housed and how many mice per cage. If you do not have many cages, it is unfortunate, but it should be clear. Please also make sure to describe the sample collection and BAL procedures more clearly.

---

## [Author Response]

Essential revisions:Addition of qPCR and/or FISH data: The authors did not show the 16S qPCR data to compare the overall bacterial load between BAL, total lung homogenate, and lung homogenate after BAL. However, such data is critical to demonstrate the difference of commensal bacteria that are reside in the airway vs. those are tissue-associated. While the authors mentioned the 16S copy is higher in tissue-derived samples than BALF samples, and the BALF samples are indistinguishable from background, it is necessary to directly show the results, with the proper normalization and control. In particular, adjacent sterile tissue (such as heart) could be used as a control for lung tissue samples. Alternatively, FISH could be performed to directly demonstrate the presence of L. murinus in the lower respiratory tract and their association with lung tissue.

This is an excellent point and indeed very few papers address the specific presence of lung resident microbiota by FISH. We used highly sensitive RNAScope technology and detected *L. murinus* and total eubacteria with specific probes in neighboring sections of SPF mouse lungs. GF mouse lung sections were used as negative controls. As shown in the new Figure 3B we found a clear and matching staining for both probes in large and medium sized airways.

In parallel we performed qPCR assays using *L murinus* specific primers and probes to quantify *L. murinus* in the lungs of SPF mice (new Figure 3A). Using a standard of genomic DNA from cultured *L. murinus* with a defined cfu we estimated the number of *L. murinus* to be around 10e7/lung. This estimate is about 100fold higher than the number of total cultured bacteria determined previously [Yildiz et al., 2018], which might be due to the presence of dead bacteria or a consequence of alternative organ processing for the two techniques.

in vivo measurements of pH and lactic acid levels: It is not clear if the proposed mechanism is physiologically relevant. Data on pH and lactic acid concentration of BALF samples from L. murinus colonized versus GF mice would be a good addition to the paper. This mechanism also seems to contradict the proteinase K experiment, which implicates a protein/peptide.

We agree with the notion that there might be more than mechanism in place, reducing replication of *S. pneumoniae* in presence of L. murinus. We compared lactate levels and pH in BALF of *L. murinus* treated SPF mice. We chose this model, since we observed here an effect on *S. pneumoniae* colonization. Lactate levels were unchanged 3 days post *L. murinus* inoculation in mock or influenza A virus pre-treated mice, which might be a consequence of quick absorption of lactic acid into the tissue or commensal bacteria. Nevertheless, we found a significant reduction of pH by 0.1 in BALF of mice treated with *L. murinus*. While this appears to be minor effect, previous publications suggest that a drop of 0.5 (assuming an overall pH of 7 in the lung), lead to 4 log10 reduction in in vitro experiments [Mazzola et al., 2003], which is clearly more dramatic than what we observed in our in vivo experiments. These data are included in the new Figure 8D and Figure 8—figure supplement 1A.

The authors should evaluate the alternative hypothesis that the impact of L. murinus is mediated through interactions with the immune system. As a first pass, the authors could perform cytokine analysis from the BAL/tissue of the animals or ideal repeat the experiment with flow cytometry.

Again, an excellent point: We performed qPCR analysis of mRNAs involved in inflammatory responses to bacteria. While we observed a mild reduction in mRNA levels for TNFa, IFNγ and IP10, these differences do not reach statistical significance. These data were included into the new Figure 8B.

The authors observe that the β diversity and 16S qPCR copy numbers of BAL and blanks (controls) were similar. This is surprising given that several other studies in mouse and humans (Charlson et al., 2012, Dickson, 2014) have clearly demonstrated significant differences in controls and BAL. The authors suggest that this indicates a high number of false positives. What do the authors think is the cause for this? Does it reflect an issue with contamination of the saline/sterile water used for BAL, their sequencing techniques, or something else?

We use as blank controls sampling tubes that are opened for a few seconds in the same biosafety cabinet, where mouse organs are samples, after this they are processed as all the other samples. Sterile PBS is aliquoted under a clean BSC for these experiments. Contamination could originate from the DNA preparation kit (which might pose a source of experimental variation between the reference above and our study) or cross contamination during library preparation (although this is rather unlikely, since the blanks are randomly included in the PCR plates.

Need to deposit sequencing data prior to resubmission. This includes the sequencing data from lung or bal of the mice from both model systems that were inoculated with L. murinus.

NGS sequences (including the new data from Figure 8—figure supplement 1C and D) are available under NCBI BioProject numbers PRJNA591377 (NGS MiSeq microbiome data).

This will provide insight into the other commensals or pathobionts that may have been are promoted or suppressed by L. murinus. It is possible that growth of other pathobionts/commensals in the lung due to L. murinus exposure may be in part or indirectly responsible for suppression of S. pneumoniae.

We performed 16S rRNA gene specific NGS on lung DNA of SPF mice and SPF mice treated with L. murinus. While we see an enhanced presence of Bacilli (as consequence of probiotics treatment) we do not see major shifts in the remaining microbial community. Accordingly, there are not striking differences in the β diversity of SPF and SPF+*L. murinus* mice. These data are now included into the new Figure 8C and 8D.

[Editors' note: further revisions were suggested prior to acceptance, as described below.]

Essential revisions:The reviewers request that multiple claims in the paper be modified or weakened, setting the stage for future studies. The appropriate caveats could be added to the results or Discussion section. This includes:It is still not clear what the overall bacterial load is in BAL, total lung homogenate, and lung homogenate after BAL by 16S qPCR. Without such data, there is not sufficient evidence to claim L. murinus is tightly associated with lung tissue.

We now provided the qPCR data for panbacterial 16S rRNA and *L. murinus* 16S rRNA for the three extraction conditions (BALF, lung after BALF and lung) from SPF housed and conventional mice (Figure 2—figure supplement 1A and Figure 3B).

Regarding the physiological relevance of lactate derived from L. murinus, the authors didn't observe any changes 3 days post L. murinus inoculation in mock or influenza A virus pre-treated mice.

This is correct. We can only speculate that the lactate might already have been taken up by the surrounding tissue or bacteria. Lactate-independent mechanisms (e.g. the reduced chain formation) acting in vivo might be an alternative explanation. We included this point into the Results and Discussion.

Cytokine levels are interesting but would require flow cytometry and additional experiments to better understand how the immune system is involved.

We actually do not think that there is a major effect on cytokine production. Statistically the reduction in cytokine mRNA levels from IAV infected mice to IAV+*L murinus* treated mice is not significant. Moreover, a reduced level of cytokines involved in antibacterial signaling would argue against an involvement of the host immune system, since we see less S. pneumonia in the lung after *L. murinus* addition.

Contamination introduced from the DNA preparation kit would also be present in lung samples.

Blank samples indicated in Figure 2A represent the contamination coming from the DNA preparation kits. We also provide a new Figure 2—figure supplement 1C showing the overall read counts from lung and blank samples.

Subsection “Lactobacillus murinus (ASF361) is a major constituent of mouse lung microbiota”: Please revise this subheading to be more specific to the current study, unless the authors meant to imply that L. murinus is a major player in the lung microbiota in all mice.

We agree that our findings potentially only apply to the chosen experimental conditions. In line with the reviewers comment we changed the subheading to:

“*Lactobacillusmurinus* (ASF361) could be a major constituent of mouse lung microbiota in laboratory settings”

We additionally included the sentences: " A limitation to these findings is that they are based on data from two laboratories within a single animal facility. Facility and vendor specific confounding effects will require to be addressed in the future."

Please discuss why germ-free mice are resistant to S. pneumoniae infection regardless of L. murinus pre-treatment.

We included the following in the Discussion:

“Notably we found that germ-free mice were more difficult to colonize with *S. pneumoniae*. This appears as a contradiction to the hypothesis that lung microbiota poses a colonization barrier toward invading bacterial pathogens. An explanation to this dilemma could be that *S. pneumoniae* colonization requires yet unknown anatomical conditions induced by commensal lung microbiota. Future tests in antibiotic treated mice might resolve this phenomenon.”

Make sure to include the appropriate statistical analysis and comparison groups in each figure.

Statistical test are now included whenever appropriate in the figures and explained in the figure legend.

Subsection “Lactobacillus murinus (ASF361) is a major constituent of mouse lung microbiota”: Show the data on 16S copy numbers based on qPCR.

We included qPCR data for 16S and *L. murinus* for BALF, lung after BAL and total lung samples (Figure 2—figure supplement 1A and Figure 3B).

Materials and methods section: It is imperative to describe how many cages, how they were housed and how many mice per cage. If you do not have many cages, it is unfortunate, but it should be clear. Please also make sure to describe the sample collection and BAL procedures more clearly.

This information is now included into Figure legend 2. The BAL procedure was described in more detail (subsection “Fluorescence *in situ* hybridization”).